# Genetic screen of the yeast environmental stress response dynamics uncovers distinct regulatory phases

Jenia Gutin ⓘ, Daphna Joseph-Strauss, Amit Sadeh, Eli Shalom & Nir Friedman* ⓘ

## Abstract

Cells respond to environmental fluctuations by regulating multiple transcriptional programs. This response can be studied by measuring the effect of environmental changes on the transcriptome or the proteome of the cell at the end of the response. However, the dynamics of the response reflect the working of the regulatory mechanisms in action. Here, we utilized a fluorescent stress reporter gene to track the dynamics of protein production in yeast responding to environmental stress. The response is modulated by changes in both the duration and rate of transcription. We probed the underlying molecular pathways controlling these two dimensions using a library of ~1,600 single- and double-mutant strains. Dissection of the effects of these mutants and the interactions between them identified distinct modulators of response duration and response rate. Using a combination of mRNA-seq and live-cell microscopy, we uncover mechanisms by which Msn2/4, Mck1, Msn5, and the cAMP/PKA pathway modulate the response of a large module of stress-induced genes in two discrete regulatory phases. Our results and analysis show that transcriptional stress response is regulated by multiple mechanisms that overlap in time and cellular location.

**Keywords** budding yeast; dynamics of transcriptional response to stress; genetic interactions; Msn2 and Msn4; signaling pathways
**Subject Category** Signal Transduction
**Mol Syst Biol. (2019) 15: e8939**

## Introduction

Cells are exposed to varying extracellular and intracellular conditions. Survival and adaptation to fluctuating environments require adjustments of the cellular transcriptional program. Some environmental changes lead to a slow and gradual transcriptional response, while others result in a rapid and transient response. The precise regulation of the transcriptional response is crucial and involves a large number of interconnected sensing, signaling, and transcription regulatory pathways. Given different environmental inputs, those regulatory networks determine which genes will be activated or repressed and shape the dynamic properties of the response. Over the years, numerous studies, performed in different systems and organisms, have characterized the sets of genes repressed and activated in response to environmental changes (Gasch *et al*, 2000; Causton *et al*, 2001; Price *et al*, 2001; Girardot *et al*, 2004; Rodriguez *et al*, 2013). In some of these systems, there is an extensive knowledge about the specific signaling pathways and transcription factors that participate in the regulation (Bahn *et al*, 2007; Zaman *et al*, 2008; Tower, 2012; Rodriguez *et al*, 2013). While we have a good understanding of the mechanisms that are necessary for transcriptional response (e.g., key factors whose inhibition abolishes a certain response), much less is known about the molecular mechanisms that modulate and shape its dynamics.

The response dynamics should satisfy two seemingly inconsistent requirements. On the one hand, the magnitude of the response should be proportional to the level of stress and thus incorporates some level of feedback. On the other hand, an ideal response would shut down transcription before actual feedback may occur, yet there is unavoidable and non-trivial delay between transcriptional events and actual consequences (e.g., synthesis of proteins and their corrective actions). In fact, there is a fitness cost for overly prolonged responses as cells stop growing and synthesize proteins that are not needed (Brauer *et al*, 2008; López-Maury *et al*, 2008; de Nadal *et al*, 2011; Paek *et al*, 2016). Thus, simple feedback loops would either be too slow in turning off the response, or involve delays in turning it on (Alon, 2006). Stated differently, response to stress can fall on a spectrum between two extremes. On one end is a fully predetermined response, where the response magnitude matches from the outset the severity of the insult. On the other end is an adaptive response initially identical for all levels of insults, and then progresses differently depending on the recovery. These can be thought of as sensing either the exact damage or the recovery to pre-stress conditions.

An attractive model system for studying transcriptional response dynamics is the yeast *Saccharomyces cerevisiae*, where environmental changes activate both condition-specific transcriptional programs and a general response program (Gasch *et al*, 2000; Causton *et al*,

School of Computer Science and Engineering and Institute of Life Sciences, The Hebrew University of Jerusalem, Jerusalem, Israel
*Corresponding author. Tel: +97 225 494557; E-mail: nir.friedman@mail.huji.ac.il

2001; Hohmann, 2002; Hohmann & Mager, 2007; Morano *et al*, 2011). This program is often referred to as the *environmental stress response* (ESR). The ESR involves hundreds of genes that can be either repressed or induced (iESR) in response to environmental stress (Gasch *et al*, 2000; Causton *et al*, 2001). The induction of the iESR genes is regulated by two paralogous transcription factors, Msn2 and Msn4 (Martínez-Pastor *et al*, 1996; Schmitt & McEntee, 1996; Gasch *et al*, 2000; Causton *et al*, 2001). Under normal conditions, these factors are located in the cytoplasm. Upon environmental change, they translocate into the nucleus, bind *stress response elements* (STREs) at promoters of iESR genes, and induce transcription (Martínez-Pastor *et al*, 1996; Görner *et al*, 1998). The activity of Msn2/4 can be regulated at different levels: expression, nuclear import, export, and degradation (Görner *et al*, 1998, 2002; Chi *et al*, 2001; Jacquet *et al*, 2003; Durchschlag *et al*, 2004; Lallet *et al*, 2006; Hao & O'Shea, 2011). It was shown that many cellular pathways, including cAMP/PKA, TOR, SNF1/AMPK, and HOG MAPK, regulate Msn2/4 activity (Görner *et al*, 1998; Santhanam *et al*, 2004; De Wever *et al*, 2005; Garmendia-Torres *et al*, 2007; Capaldi *et al*, 2008; Gutin *et al*, 2015). However, the contribution of each pathway varies in different conditions (Sadeh *et al*, 2011; Gutin *et al*, 2015). Moreover, different environmental conditions and varying severity of the same condition can induce responses with dramatically different dynamics in terms of intensity and duration (Gasch *et al*, 2000; O'Rourke & Herskowitz, 2004; Hansen & O'Shea, 2013).

Here, we examined the dynamics of iESR induction following environmental insults. As a phenotype for this experiment, we measured the dynamics of accumulation of Hsp12-GFP, a well-established reporter for Msn2/4 activity (Martínez-Pastor *et al*, 1996; Gutin *et al*, 2015), in response to a gradient of osmotic stress levels. We observed a complex response where both the intensity and the duration are modulated. To investigate what are the regulatory mechanisms involved in this complex response, we used genetic perturbations—single knockouts and double knockouts—to uncover the contribution of individual genes to the dynamics of the iESR response and to identify epistatic interactions between multiple genes. We uncover key regulatory pathways that determine the duration and intensity of *HSP12* transcriptional response. Using mRNA-seq, we show that the same mechanisms apply to most of the iESR genes, yet achieve multiple dynamic behaviors of different genes with the shared signaling pathway. We show that this is due, in part, to multiple regulatory phases that depend on different signaling mechanisms.

## Results

### Parameterization of response dynamics reveals differences along two different dimensions

To study iESR induction, we extended prior efforts (Martínez-Pastor *et al*, 1996; Sadeh *et al*, 2011; Gutin *et al*, 2015) using GFP-tagged *HSP12* as a highly sensitive stress-responsive reporter. Hsp12 is a small heat-shock protein produced at a high rate following a variety of stresses. Hsp12-GFP response is indicative of the level of stress and the condition of the cell (Gutin *et al*, 2015) and thus provides a good proxy for iESR induction.

To better understand how yeast respond to varying levels of insults, we used flow cytometry to measure the induction of the HSP12-GFP fluorescent reporter construct following stimulus (Materials and Methods). As a model stress condition, we use KCl-induced osmotic stress. This condition has been extensively studied (Hohmann, 2002). Briefly, yeast exposed to increased osmotic levels go through a phase of immediate shock, followed by acclimation stage that involves massive transcriptional changes, and finally after adaptation continue to grow in the new environment. After adaptation, gene expression levels are close to the original pre-stress levels (Gasch *et al*, 2000; Hohmann, 2002).

In our hands, after exposing yeast to different levels of osmotic stress (0.15–0.8 M KCl), we observe a large dynamic range of Hsp12-GFP induction (Figs 1A and EV1A). This response is dependent on several molecular mechanisms, as we find a range of responses to the same level of stress (0.4 M KCl) in different mutant strains (Fig 1B). We sought to compare these responses and quantify their key properties. Naively, we can directly compare the accumulation curve in each response at preset times (Sadeh *et al*, 2011; Gutin *et al*, 2015). Such a comparison depends on the exact timing of the measurement (Fig 1A and B). Alternatively, we might compute the distance between the response curves (e.g., Euclidean distance); however, this measure does not distinguish the timing of the differences and their sources.

Instead, we aimed for a model-based characterization of the response. A simple description of the induction process is one where the protein accumulation is a function of RNA levels, which are in turn determined by a transient window of active transcription (Fig 1C) and a constant RNA degradation rate. This model is supported by the fast response to stress followed by acclimation to the new conditions (Hohmann, 2002). While over-simplified, this model captures the major variables in transient response. Indeed, with few parameters the model can fit the empirically observed responses to a range of insults, across a variety of genetic backgrounds (Fig 1A and B) and captures > 99% of the variance in the response (Materials and Methods, Fig EV1B and C).

This representation allows us to summarize response curves with three straightforward parameters (Fig 1C)—response onset time ($T_{on}$), response end time ($T_{off}$), and production rate ($\alpha$). We note that production rate includes both transcriptional and translational processes as we cannot distinguish the two with this experimental readout.

After extracting these parameters, we now can compare response curves measured at different time points, at different temporal resolutions, across different experimental settings and different readout assays (FACS, microscope, mRNA-seq). For example, we can represent the responses of different KCl stress levels and different mutant strains in the parameter space (Fig 1D). Examining the response to increasing gradient of KCl levels, we see as expected that higher levels of KCl result in increased production of Hsp12-GFP (Fig EV1D). However, these levels of production exhibit different dynamics (Fig 1A). The parameterization allows us to understand these differences. At KCl concentrations below 0.2 M, increasing the stress level leads to faster onset of the response (due to lack of synchronization between cells in low levels of stress (Pelet *et al*, 2011)). However, increasing KCl levels beyond 0.2 M leads to an increasing delay in the response (Fig 1E). This delay is consistent with the physiology of the osmotic stress response, which requires

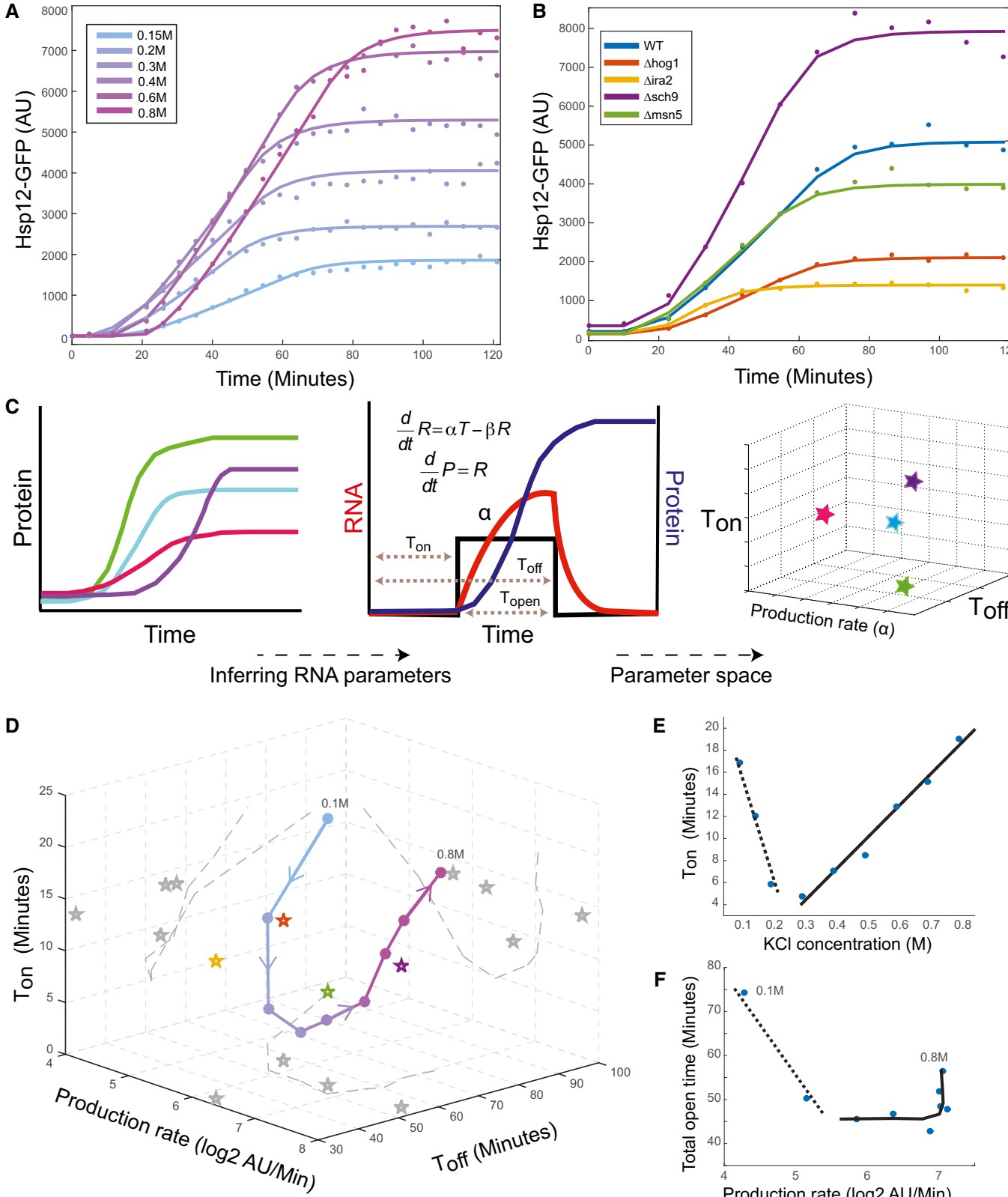

**Figure 1.**

◄

**Figure 1.  A parametric view of dynamic transcriptional response.**

A   Hsp12-GFP accumulation (y-axis) versus time (x-axis) in response to different levels of KCl stress. Points reflect the measured values, and lines correspond to best model fit.

B   Similar to (A), in different genetic backgrounds at 0.4 M KCl.

C   A scheme of parametric decomposition. (i) Start from protein abundance measurements in different time points following stress. (ii) Fit RNA parameters based on a simplified model for transient transcriptional response (Weiner et al, 2012) and (iii) compare different conditions/perturbations in parameter space.

D   Representation of the experiments shown in (B, C) in parameter space. Connected line shows the progression in the response to increasing KCl levels (see B).

E   The inferred $t_{on}$ parameter (y-axis) versus KCl concentration (x-axis) of the KCl gradient experiment (see B).

F   The inferred total open time $- t_{off} - t_{on}$ (y-axis) versus the inferred production rate (x-axis) of the KCl gradient experiment (see B).

re-establishment of proper nuclear salt level and cellular volume prior to productive transcription (Hohmann, 2002; Proft & Struhl, 2004; Muzzey et al, 2009). Moreover, between 0.15 and 0.4 M KCl, the total time of response ($T_{off}$ - $T_{on}$) is roughly constant, but the production rate increases gradually. This behavior suggests that cells can differentiate the degree of damage across this range (damage sensing regime). However, at concentrations of 0.5 M KCl and higher, we see a roughly constant production rate with increasing response time, suggesting that the cells switch to an adaptive regime, where they induce the same stress production rate and achieve higher levels of proteins by increasing the duration of the transcription window (Fig 1F). As we show below, this level of response is far from saturating the production rate.

These observations suggest that the dynamic parameters are regulated in at least two regimes (low stress and high stress), consistent with prior observations on the stress sensing pathway (O'Rourke & Herskowitz, 2004; Gutin et al, 2015). Moreover, we see that the response varies in the two dimensions (response duration and response rate) which can be regulated separately.

### Partially decoupled pathways regulate dynamic response parameters

To determine what mechanisms modulate these parameters, we performed a genetic screen. Using automated flow cytometer setup, we measured Hsp12-GFP dynamics at 6 time points following shift to 0.4 M KCl media (Materials and Methods). We performed this analysis on a collection of 68 mutant strains including mutants of genes in the HOG MAPK pathway, TOR pathway, cAMP/PKA pathway, and other pathways related to the yeast stress response (Gutin et al, 2015). These perturbations led to a range of response curves (e.g., Fig 1B) which we summarize by each mutant's impact on the parameters described previously (total GFP produced, response onset time, duration of transcription window, and production rate; Fig 2A).

We observe that many of the perturbations have dramatic effects mainly on a single parameter (Fig 2A). For example, deletions of HOG1 and PBS2 genes (encoding the MAPK and MAPKK of the HOG signaling pathway) significantly reduce Hsp12-GFP production rate while having a milder effect on the duration of production window. In contrast, deletion of PDE1 and PDE2 (two cAMP phosphodiesterases) affects the duration more than the production rate. More broadly, we find 19 mutants that mainly affect the response duration, 11 that mainly affect the production rate, and 4 mutants that affect both. Satisfyingly, genes in the same pathway often have a similar parameter profile (e.g., cAMP pathway components PDE1, PDE2, and IRA2; PBS2 and HOG1; GIS1 and RIM15; elongator components ELP4 and ELP6). This analysis allows us to determine

which genes regulate response timing, which regulate production rate, and which influence both (Fig 2B).

### Response dynamics are mostly dictated at the mRNA level

Our analysis until this point is based on Hsp12-GFP levels. Thus, we cannot distinguish between transcriptional and translational effects of the genetic perturbations. To examine this question directly, we used 3′ mRNA-seq (Materials and Methods) to evaluate mRNA levels in a time course following 0.4 M KCl treatment in 16 mutant strains (single replicate per strain, three for WT), chosen to represent a range of parameter profiles (Fig 2C). Examining HSP12 mRNA levels throughout the response (Fig 2D), we see a rapid increase in RNA levels followed by their decrease as the yeast acclimates to the new conditions (Gasch et al, 2000). As expected, different mutant strains had a range of HSP12 responses, with barely noticeable induction in Δhog1, Δira2, and Δgal11, and a stronger than WT induction in Δmsc1 and Δeaf7 (Fig 2D).

We assumed until now that protein production should be proportional to the integral of mRNA levels throughout the response. To test that, we contrasted mRNA levels to protein production levels. Indeed, the integral of HSP12 mRNA is strongly correlated with maximal Hsp12-GFP levels ($R^2$ = 0.82, Fig 2E). A noticeable exception to this trend is the deletion of the SCH9 gene, encoding for a Tor1 target kinase that regulates ribosome biogenesis and translation initiation (Huber et al, 2009). In Δsch9, the observed protein levels are double of the expected amount based on mRNA levels, suggesting that Δsch9 effect has a major translational component.

Having observed agreement between total levels of protein and mRNA, we next asked whether the dynamics of mRNA induction match the ones inferred from Hsp12-GFP data. Using the same model (Fig 1C, middle panel, red curve), we extracted the dynamic parameters from the HSP12 mRNA data. Comparing the parameters estimated from mRNA profiles and Hsp12-GFP profiles, we see good agreement (Fig 2F and G). The total open time parameter estimated from protein data tends to be higher than the one estimated from mRNA data, probably due to the effect of maturation time on Hsp12-GFP measurements. As expected, Δsch9 is an outlier in production rate, but not in total open time, supporting the claim that this mutant affects the translation of HSP12 mRNA. In general, we conclude that transcriptional parameters estimated from protein levels are a good proxy to study transcriptional response.

### Population-level response is representative of individual cells

Our analysis up to this point is based on the population mean (either of RNA or GFP) during the response. However, it is possible that response variability between individual cells will skew

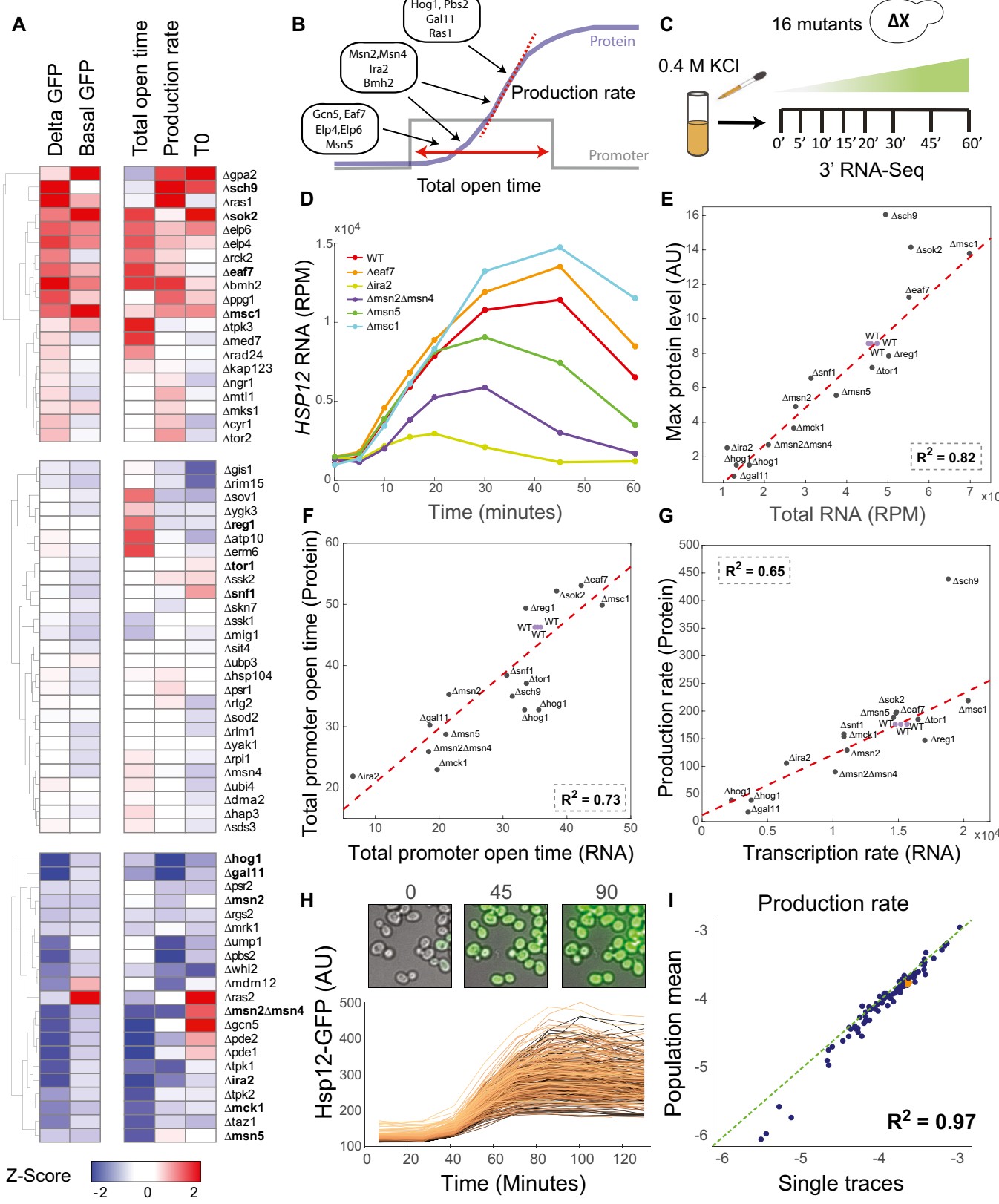

Figure 2.

◄

**Figure 2.  Partially decoupled regulation of transcriptional dynamic parameters inferred from protein and RNA measurements.**

A  Summary of Hsp12-GFP dynamics single-mutant screen. Sixty-eight mutants were divided into three groups: increased/decreased/did not change the total GFP produced. Each group was independently hierarchically clustered. The values shown are the Z-scores of each parameter: $t_{on}$ —response onset time; production rate; total open time—length of transcriptional window ($t_{off}-t_{on}$); basal GFP—GFP levels measured before exposure to stress; delta GFP—amount of GFP produced throughout the experiment (max GFP - basal GFP).

B  The dynamic parameters are regulated by partially independent mechanisms. Mutant strains can be roughly divided into three groups: affecting total open time, affecting production rate, and affecting both.

C  Outline of mRNA-seq experiment. Sixteen yeast strains were grown to mid-log phase, exposed to 0.4 M KCl, and sampled in the indicated time points after exposure to stress.

D  mRNA levels of *HSP12* (y-axis) as a function of time (x-axis) in six representative strains.

E  Comparison of RNA levels to protein levels. X-axis is the sum over the *HSP12* mRNA counts (see D). Y-axis is the maximum over the Hsp12-GFP values. The red line indicates the best linear fit.

F  Comparing the total open time parameter inferred from mRNA data (x-axis) to the one inferred from protein data (y-axis). The red line indicates the best linear fit.

G  The same as (F) for the production/transcription rates. The red line indicates the best linear fit.

H  Time-lapse microscopy of Hsp12-GFP in response to stress (0.4 M KCl). Examples of raw images (top) and traces of individual cells (bottom). Traces are colored according to their basal GFP level.

I  Scatter plot showing the production rate estimated from single traces versus the production rate estimated from the population average. Each point represents a different yeast strain.

estimated parameters (Fig EV2A). Our flow cytometry assay samples the population at each time point and thus provides an estimate of variability at each time point. Such measurements, however, do not provide single-cell dynamics as any given individual cell is only observed at a single time point.

To directly assay the transcriptional dynamics in single cells, we performed live-cell microscopy experiment in 96 mutant strains with HSP12-GFP reporter in two repeats (Materials and Methods). Briefly, mid-log growing cells were adhered to a glass-bottom multiwell plate, then exposed to stress (0.4 M KCl), and observed for ~180 min with imaging of brightfield and GFP every ~15 min (Fig 2H). Image analysis recovered the trajectory of GFP in individual cells visualized during the experiments. From each population, we recovered 200–1,000 high-quality single-cell traces (Fig 2H, Materials and Methods). For each strain, we estimate the model parameters from the population mean (Fig EV2B) and from each one of the traces (Fig 2H), resulting in a distribution of points in parameter space that represent variability in dynamics of genetically identical cells (Fig EV2C–E).

Comparing the estimate of parameters from the population mean to the mean of the distribution over parameters of individual cell traces, we observe an excellent agreement across all 96 strains (Figs 2I and EV2F). Importantly, estimates based on population means reflected differences between strains that are smaller than the variability within the population of each strain. Together, these results suggest that in most cases population-level parameters represent the dynamics in individual cells.

### Dynamics of stress response differentiate between identical accumulated outputs

Having gained confidence in the reporter assay and the parameterization of dynamics, we wanted to further understand the molecular mechanisms behind this response. We moved on to quantify Hsp12-GFP dynamics in ~1,600 double-mutant strains which we have previously generated (Gutin *et al*, 2015). We measured GFP levels at six time points after exposure to 0.4 M KCl (Fig 3A, Materials and Methods) and extracted the dynamic parameters for each curve (Materials and Methods, Datasets EV1 and EV3 (for raw data)). The estimated model parameters were in good agreement between biological repeats (Fig EV3A) and explain > 99% of the signal

(Fig EV3B). Examining the fitted parameter space, we observe a wide range of parameter combinations demonstrating a varied space of possible responses (Fig 3B). Importantly, the same total GFP levels at the end of the response can be the result of significantly different dynamics (Figs 3B and C, and EV3C). Thus, examining the response dynamics uncovers additional distinctions between perturbations that are undetected when measuring only levels of total GFP at the end of the response.

### Epistatic analysis identifies regulatory interactions in modulating response dynamics

We wondered whether these finer differences provide additional information on the underlying pathways. Examining the effect of double mutants in the dynamic setting, we observe that some mutants have dominant effects in one or in both parameters (Fig 3D). For example, deletion of *HOG1* leads to reduced production rate in most combinations with other mutants (Fig 3D). On the other hand, deletion of *SCH9* gene increases Hsp12-GFP production rate in most combinations. The deletion of *ELP6* gene has a large effect on response duration and does not affect the production rate (Fig 3D). In some cases, a mutant can have dominant effects in both parameters. For example, the deletion of *IRA2* gene results in shorter and weaker response in most combinations with other mutants (Fig 3D).

A powerful way of analyzing genetic interactions is in terms of epistasis. We say that a deletion of X is *epistatic* over a deletion of Y when ΔXΔY has the phenotype of ΔX and is different from the phenotype of ΔY (Phillips, 2008). Previously, we analyzed epistatic relations based on total GFP produced after stress in the same strains (Gutin *et al*, 2015). Reasoning that dynamics of responses provides richer information, we defined a test for epistatic relations between pairs of genes (Materials and Methods) and applied it to total GFP produced, production rate, and response duration (Dataset EV2).

For example, among Δhog1 strains (Fig 3E) we observe 34 double mutants where the deletion of *HOG1* gene reverted the production rate in a single mutant to a Δhog1-like production rate in the double mutant (e.g., Δhog1Δrck2). Similarly, Δhog1 is epistatic over many genes in terms of response duration (Fig 3F). Interestingly, an epistasis interaction in one parameter does not imply an epistasis in the other. For example, Δhog1 is epistatic over Δreg1 in

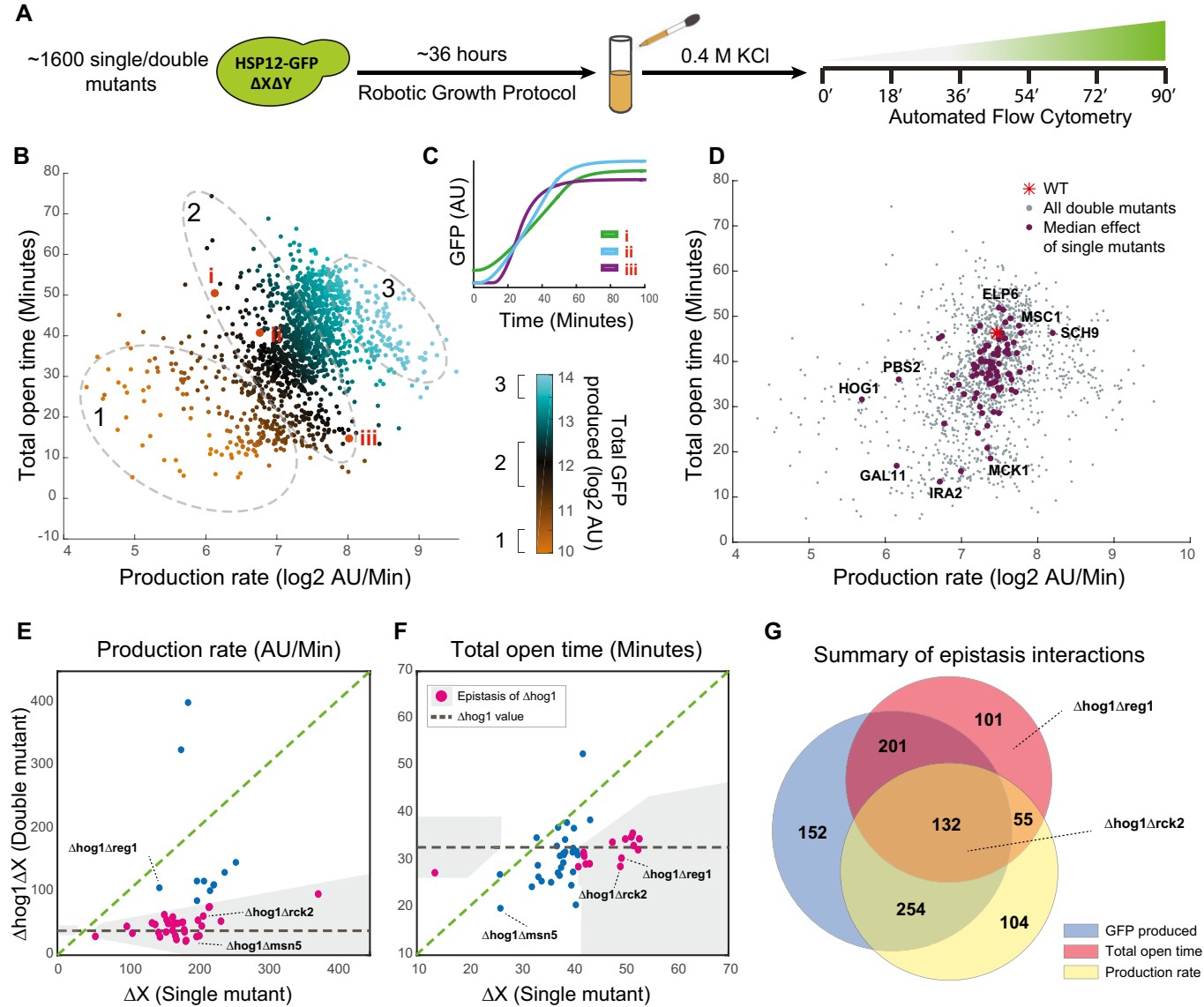

**Figure 3. Double-mutant screen for regulators of dynamic parameters.**

A   Double-mutant library, containing ~1,600 strains with Hsp12-GFP reporter, was screened using automated flow cytometry in six time points following exposure to stress (0.4 M KCl).

B   Estimated parameters (median over repeats) of all ~1,600 double-mutant strains. The color represents the total Hsp12-GFP produced. The same amount of GFP can be produced with multiple parameter combinations. The ellipses illustrate iso-expression areas in the graph.

C   Three examples of individual strains (marked in B) with nearly the same total expression but dramatically different dynamics.

D   Illustration of the genetic interaction trends of individual mutants. For each single mutant, we plot the median of all the double mutants that contain it (purple dots). The gray dots are all individual double-mutant strains (B).

E, F   Illustration of the interactions of Δhog1 in the production rate parameter (E) and the total open time parameter (F). Shown the production rate of the single mutant (x-axis) versus the production rate of the double mutant with Δhog1 (y-axis). The gray line marks the Δhog1 single-mutant levels. Points close to the gray line are ones where the value of the double mutant is close to the value of Δhog1 and defined as epistasis of Δhog1 (pink dots).

G   Venn diagram comparing the number of epistatic pairs detected in various phenotypes.

terms of response duration but not in terms of production rate. In contrast, in the case of Δrck2 we observe epistasis of Δhog1 in both parameters. Importantly, limiting the epistasis analysis only to the total GFP produced parameter would not recover the epistasis of Δhog1 over Δreg1 (Fig EV3D).

Our epistatic analysis shows several trends. First, a majority of the epistatic interactions observed from total GFP (587/739) are

detected in epistatic analysis of one or two of the dynamic parameters (Fig 3G). In most of the cases (510/587), the direction of epistasis is in agreement (Fig EV3E). Second, in the cases where there is an agreement, the refined analysis identifies the main regulatory aspect of epistasis in the specific pair (e.g., production rate in the case of Δhog1Δmsn5). Thirdly, epistatic analysis of individual parameters identified 205 new epistatic interactions, such as

Δhog1Δreg1, that could not be inferred by analysis of the total GFP (Fig 3E–G). These involve epistasis in one parameter but not the other, leading to seemingly non-epistatic interaction in terms of the total GFP produced. Finally, there is a small number (55) of pairs with epistatic interactions in both parameters that are not detectable in total GFP. Many of these (31/55) involve incoherent interactions —first mutant is epistatic over the second in response time, and the second is epistatic over the first in production rate (Fig EV3E).

We conclude that genetic analysis of individual aspects of the response dynamics is highly informative and can uncover multiple epistatic interactions that are hidden when examining only a single measure at the end of the response (e.g., total GFP produced).

## MCK1 gene deletion significantly shortens the Msn2/4-dependent transcriptional response

Using these epistatic maps, we returned to ask what genes regulate the extension of the response in high stress conditions (Fig 1F). Examining mutants that had strong effects on the total open time parameter but had a small effect on the production rate (Fig 3D), we decided to focus on *MCK1*. *MCK1* encodes for a protein kinase related to the mammalian GSK-3 (Neigeborn & Mitchell, 1991; Bianchi *et al*, 1993). Examining the epistasis interactions of Δmck1, we see that their vast majority are in the total open time parameter (Fig 4A and B), suggesting a central role of Mck1 in determining the length of the transcriptional window.

Mck1 was previously linked to the ESR and has been suggested to regulate Msn2/4 activity, either directly or indirectly (Hirata *et al*, 2003; Sadeh *et al*, 2011; Gutin *et al*, 2015). To check whether the deletion of *MCK1* gene completely abolishes the activity of Msn2/4, we compared the dynamics of *HSP12* mRNA production in the Δmck1 and Δmsn2Δmsn4 strains (Fig 4C upper panel). In general, both strains significantly reduce the total RNA level and the response time. However, at the first 10–15 min of the response, the Δmck1 strain accumulates mRNA in a rate similar to WT, consistent with our observations using Hsp12-GFP. These results show that there is residual activity of Msn2/4 in Δmck1 strain.

To characterize this residual activity, we approximated the Msn2/4-dependent production at each time point (Fig 4C lower panel, Fig EV4A, Materials and Methods). In WT strain, we see two significant Msn2/4-dependent production peaks. However, in Δmck1 strain, there is only one such peak, 10 min after the exposure to stress, while the rest of the response is similar to Δmsn2Δmsn4 (Msn2/4 independent). This suggests that at the beginning of the response, Msn2/4 is active in the absence of Mck1. However, this activity is prematurely abolished, raising the hypothesis that Mck1 is involved in prolonging the duration of Msn2/4-dependent *HSP12* transcription.

## Mck1 has a global effect on the general stress response

We reasoned that this effect might be either specific to *HSP12*, or a global effect on the iESR genes. We thus examined the mRNA profiles of 230 stress-induced genes during response to 0.4 M KCl (Fig 2C) and calculated the integral over the mRNA of each gene throughout the response. Comparing the integrals of WT and Δmck1 (Fig 4D), we see that the deletion of *MCK1* reduces the total response levels of most stress-induced genes. The group of genes

that are Mck1-independent consists mostly of Msn2/4-independent genes (Fig 4D—red dots, Fig EV4B), again suggesting that the effect of Mck1 is through regulation of Msn2/4.

Next, we asked whether the dynamic response of all Msn2/4-dependent genes is affected similarly to *HSP12* by the deletion of *MCK1* gene. We noticed that for some Msn2/4-dependent genes, both Δmck1 and Δmsn2Δmsn4 cause the same effect, while in others the effect of Δmsn2Δmsn4 is much stronger (Fig EV4B). To further understand these differences, we sorted the genes by the onset time of the response (Fig 4E). Interestingly, genes with early onset after response to stress show higher induction in the Δmck1 strain in comparison with the late-onset genes (Fig 4E and F). Examining the Msn2/4-dependent production profiles of the early-onset genes in the Δmck1 strain (Fig 4F upper panel, Fig EV4C), we see that similarly to *HSP12*, there is a significant Msn2/4-dependent transcription of these genes 10 min after the exposure to stress. In contrast, in the late-onset genes we do not see Msn2/4-dependent transcription (Fig 4F lower panel, Fig EV4C).

These observations can be explained by a simple mechanism in which there are two time windows of Msn2/4 activity. The first one is Mck1-independent, and the second one is Mck1-dependent. Genes whose onset time is within the first window will show Msn2/4-dependent transcription in Δmck1 strain and other genes will not. We conclude that Mck1 acts in a global manner to prolong the duration of Msn2/4 activity in response to stress.

## Mck1 influences response duration through a secondary transcriptional wave

How does Mck1 achieve this global effect? To obtain mechanistic insight into Mck1 function in the context of stress response pathways, we turned again to our analysis of epistatic interactions affecting stress response dynamics (Fig 3). The deletion of *MCK1* gene is epistatic over many other mutants in terms of total open time, whereas only a handful of mutants were epistatic over Δmck1. Intriguingly, one of them, Δmsn5, reversed the phenotype of Δmck1, increasing the total open time parameter on Δmck1 background (Fig 4B). This epistasis interaction suggests that Msn5 acts downstream of Mck1 in the regulation of the response time.

Msn5 is a karyopherin that is involved in nuclear import and export (Yoshida & Blobel, 2001). Specifically, it was shown that Msn5 exports Msn2 from the nucleus (Chi *et al*, 2001; Görner *et al*, 2002). This finding led us to hypothesize that Mck1 potentially regulates the nuclear localization of Msn2/4. Upon stress, Msn2/4 are rapidly imported to the nucleus where they affect their target genes, and with the resolution of the stress, they regain cytoplasmic localization (Görner *et al*, 1998).

To better explore the extent nuclear localization is regulated by Mck1, we used microscopy to track Msn2-GFP localization following exposure to osmotic stress, as described previously (Gutin *et al*, 2015), but with higher temporal resolution (Figs 5A and EV5A, Movie EV1, Materials and Methods). In WT strain immediately after the insult, there is a rapid onset of nuclear localization that lasts ~20 min. Subsequently, there is an additional wave of nuclear import (Jacquet *et al*, 2003; Petrenko *et al*, 2013). This localization pattern is strikingly consistent with the two Msn2/4-dependent production periods (Fig 4C and F). Examining individual cell traces (Fig EV5B) uncovers additional properties of the localization

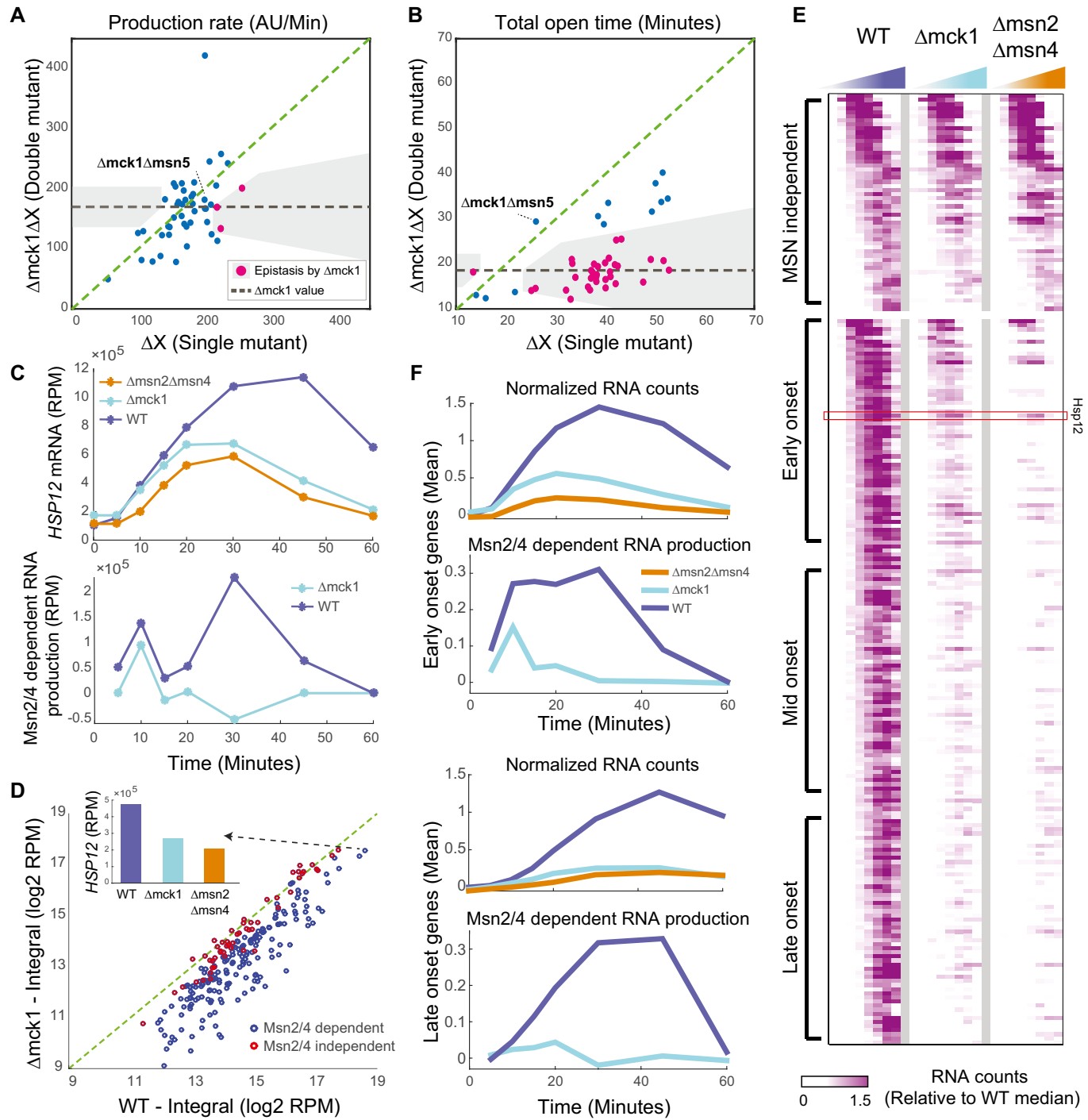

**Figure 4. MCK1 is necessary for secondary wave of Msn2/4 activity.**

A   Epistatic interactions of Δmck1 in the production rate parameter (see Fig 3E).

B   Epistatic interactions of Δmck1 in the total open time parameter (see Fig 3E).

C   Upper: *HSP12* mRNA in response to osmotic stress (0.4 M KCl). Lower: estimate of Msn2/4-dependent production of *HSP12* mRNA over time (difference between WT and Δmsn2Δmsn4 production profiles, Fig EV4A, Materials and Methods).

D   Δmck1 effect on the response of all stress genes. The integral over mRNA values in WT strain (x-axis) versus the integral in the Δmck1 strain (Y values). The inset shows the integral for the *HSP12* gene.

E   Heatmap showing expression of stress-responsive genes in WT and two mutants over eight time points (see Fig 2C). Each gene is presented relative to its pre-stress level and normalized by the median over WT values to visualize the dynamic range in all genes. The genes are separated into MSN-dependent and MSN-independent genes (Fig EV4B). The MSN-dependent genes are sorted based on the onset time to three groups (early, average, and late onset).

F   The behavior of early- and late-onset genes. Similar to (C) but showing the average over the genes in each group.

pattern. While the initial nuclear import is by and large a long continuous event (one entry and one exit), the later nuclear imports are in shorter pulses of multiple entries (Fig 5A, lower panel, and Fig EV5B). The initial import is synchronized by the stress event, while the later imports are unsynchronized among cells and thus obscured in a population-level summary.

In contrast to WT, in Δmsn5 we observe constant nuclear localization of Msn2, consistent with the prediction, while in Δira2, where the cAMP/PKA pathway is constitutively repressing Msn2 import, we see a much shorter period of nuclear localization in fewer cells. In Δmck1 strain, the initial pattern of nuclear localization is similar to the one we observe in WT. However, the second, unsynchronized, wave of nuclear import is by and large missing. This shorter response is consistent with the reduction of response duration of Hsp12-GFP in Δmck1 and the difference between early- and late-onset stress genes (Fig 4E and F).

These observations suggest that Mck1 is not required for nuclear import of Msn2/4 immediately after the stress, consistent with earlier studies (Hirata et al, 2003). Instead, Mck1 is required for mounting a second wave of Msn2 activity and thereby prolongs the duration of Msn2/4 nuclear localization.

## Nuclear export of Msn2 is necessary for efficient progression of the stress response

The deletion of MSN5 partially rescues the effect of Δmck1 (Fig 4B) through constitutive nuclear localization of Msn2. However, in Δmsn5 strain the response to stress is weaker compared to WT (Fig 5B upper panel), even though Msn2 is nuclear throughout the whole response (Fig 5A). This observation is consistent with previous results showing that nuclear localization of Msn2 is necessary but not sufficient for activity (Estruch, 2000; Durchschlag et al, 2004; Boy-Marcotte et al, 2006). The parameterization of the response allows us to see that the defect in Δmsn5 is only at the total open time parameter and not in the rate of the response (Fig 2A). Examining the Msn2/4-dependent production profile in Δmsn5 strain (Fig 5B lower panel), we see that across the first ~20 min of the response, corresponding to the first nuclear import wave (Fig 5A), Δmsn5 is similar to WT. However, the second peak of Msn2 activity is much weaker. We observe the same pattern in all Msn2/4-dependent genes (Figs 5C and EV5C).

Together, these results show that in the initial response stage, the transcriptional potential of Msn2/4 in Δmsn5 strain is similar to (or even higher than) WT strain, which is consistent with Msn2/4 availability in the nucleus. However, given the inactivity of nuclear Msn2 later in the response in Δmsn5 strain, we deduce that nuclear export (in an Msn5-dependent manner) is necessary for the induction of another productive transcriptional activation wave.

## Two separate response phases with different behavior and regulation of Msn2/4 activity

Summarizing our results, we observe two distinct phases of Msn2/4 activity during stress response. An initial acute activity followed by secondary longer response (Fig 5D). The results above suggest that there are different mechanisms involved in these two phases. During the initiation phase, there is a long pulse of Msn2 nuclear import which is highly synchronized between cells. Following this

initial burst of Msn2 import cells enter the "progression phase", involving asynchronous periodic oscillations of nuclear Msn2 (Fig 5A lower panel, Fig EV5B).

In the initiation phase, activation of Msn2/4 includes PKA-dependent nuclear import. However, this activation clearly also occurs in the presence of high pre-stress Msn2/4 nuclear levels (as in the Δmsn5 strain). Although it is possible that the transcriptional activity in Δmsn5 strain is driven by residual Msn2/4 in the cytoplasm imported into the nucleus, this is unlikely given the magnitude of the early activity being similar or stronger than in WT (Fig 5C). This suggests that the activation of nuclear Msn2/4 can be mediated by a nuclear factor independently of nuclear import. The most likely candidate is the MAPK Hog1, which is activated and imported to the nucleus upon stress (Ferrigno et al, 1998) and activates Msn2/4 (Rep et al, 2000; Capaldi et al, 2008; Gutin et al, 2015). Consistent with this logic, Δhog1 is epistatic over Δmsn5 in production rate, but Δmsn5 is epistatic over Δhog1 in response duration (Fig 3E and F).

The progression phase requires additional mechanisms. The short pulses of Msn2 nuclear localization are consistent with the pulsatile Msn2/4 localization documented in other conditions (Jacquet et al, 2003; Garmendia-Torres et al, 2007; Hao & O'Shea, 2011; Petrenko et al, 2013). The observations of shorter response and nuclear localization in Δmck1, the shorter period of activity of Msn2/4 in Δmsn5, and the epistasis of Δmsn5 over Δmck1 suggest the progression requires (i) nuclear export and (ii) re-import partially by Mck1 activity.

The emerging picture is that at the end of the initiation phase, one or more mechanisms deactivate Msn2/4. Once nuclear activators of Msn2/4 (e.g., Hog1) are deactivated and repressors of Msn2/4 (e.g., PKA) are activated, we observe a sharp drop in Msn2/4-dependent transcription. Studies of Msn2/4 localization results, including ours, suggest that in WT inactivation of Msn2/4 results in rapid nuclear export. In Δmsn5, where export is blocked, Msn2/4 are accumulated within the nucleus in an inactive form, degraded over time (Durchschlag et al, 2004), or both. The observations of a gradual decline in transcriptional activity in Δmsn5 suggest that these potential mechanisms are decoupled from nuclear export. However, we cannot rule out a scenario where targeting Msn2/4 for nuclear export involves blocking its transcriptional capacity, resulting in nuclear inactive Msn2/4 (Boy-Marcotte et al, 2006) that cannot be exported in the absence of Msn5.

How is Mck1 involved in extending Msn2/4 activity? Is this a direct or indirect effect? An earlier study (Hirata et al, 2003) has shown that stress-induced changes in Msn2 phosphorylation state do not require Mck1 (nor its paralogs). Moreover, using Co-IP these authors did not detect physical interactions between Msn2 and Mck1, either before or after stress induction. These observations indicate that this is more likely an indirect effect.

Assuming that this indirect effect, what are the potential mechanisms of activity? The results above suggest that this activity targets cytoplasmic Msn2/4. Without Mck1, Msn2/4 is not re-imported to the nucleus after the first nuclear phase. One possible mechanism is through the global cellular state. Mck1 might be required for maintaining the "stress" state of the cell. When it is missing, the cell resets the "stress" state more quickly and shuts down pathways that activate Msn2/4 and re-import it to the nucleus. Alternatively, Mck1 might influence activators (e.g., HOG pathway) or repressors (e.g., PKA pathway) of Msn2/4. Indeed, there are several connections

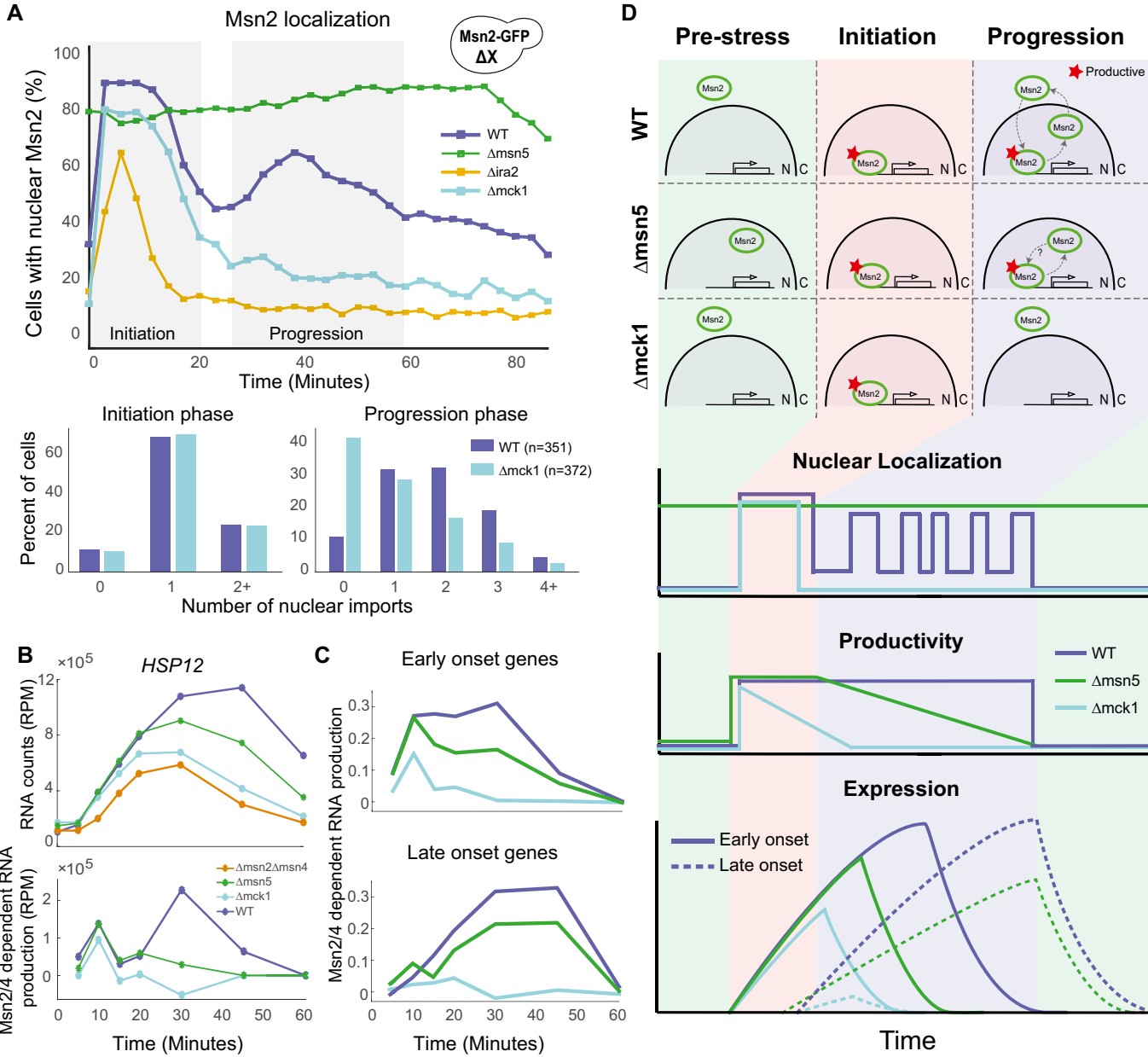

**Figure 5. Two regulatory phases in Msn2 response.**

A  Msn2-GFP localization following stress (0.4 M KCl) in selected strains. Top: the percentage of nuclear cells (*y*-axis) over time (*x*-axis). Bottom: count of the number of nuclear import events in live-cell time-lapse microscopy at the two relevant periods.

B  Same data as in Fig 4C, with the addition of the Δmsn5 strain.

C  Same data as in Fig 4F, with the addition of the Δmsn5 strain.

D  Top: Schematic model of Msn2 localization in different genetic backgrounds and phases of the response. Bottom: summary of our observations in terms of nuclear localization, Msn2 activity, and the effect on gene expression.

between Mck1 and PKA pathway: Mck1 is involved in localization of Bcy1, a regulatory subunit of PKA (Griffioen *et al*, 2003); and Mck1 was shown to repress Tpk1, one of the three PKA kinases, activity (Rayner *et al*, 2002). The effect of Mck1 activity enables the licensing of Msn2/4 for repeated nuclear import and activation at later stages of the stress response. The study of these alternative hypotheses is beyond the scope of this paper.

# Discussion

Our starting question was how cells tune response levels to stress severity. Characterizing the response dynamics to osmotic stress in budding yeast, we observe differences in dynamic response profiles to varying levels of osmotic stress (Fig 1). In low-mid stress levels, the production rate increases with more severe stress. At levels

above 0.4 M KCl, the production rate is roughly constant, and the production rate increases with the stress severity. To understand what mechanisms are involved in this phenomenon, we used a genetic approach. We evaluated the dynamics of a fluorescent reporter induction in 68 single-mutant and ~1,600 double-mutant strains. We validated the main findings by detailed mRNA-seq time course in 16 mutant strains and time-lapse microscopy in 96 mutant strains. Our analysis of this dataset reconfirmed multiple observations made in previous studies and uncovered new insights on additional modes of action of these pathways.

### Parametric representation of response dynamics

Our strategy for analyzing multiple time-course experiments is based on extracting few parameters that capture the salient features of the response dynamics. Our parametric model is based on a simplified description of transient response and has few free parameters. This choice was based in part on the nature of our measurements, where we had to strike a balance between the number of observations per strain and the number of assayed strains. In addition, we also observed that this simple model captured response dynamics in experiments with denser temporal resolution (Fig 1).

This parametric representation enabled integration of results from multiple experiments, including ones with different temporal resolution. Moreover, it allowed us to compare results from different experimental assays (flow cytometry, live-cell microscopy, and mRNA sequencing).

The parameters of the model correspond to the basic underlying biological determinants of the response—the time of response onset, the rate of transcription/translation, and the end of the response. While these determinants can involve many parts of the system (e.g., sensing, signaling, nuclear import, chromatin remodeling, and transcription initiation), the coarse resolution allowed us to match the general features with our limited observations. The decomposition along these categories allowed us to zoom in on different types of regulatory mechanisms that modulate the response.

### Decoupled regulation of time and rate parameters

Examining the effects of multiple conditions and genetic perturbations showed that many of these involved either duration of response or production rate. This (partial) decoupling further supports the choice of parameterization. Some of the pathways that affect one parameter only are expected. For example, the MAPK signaling pathway (Pbs2, Hog1) affects production rate. This pathway is involved in amplifying the activity of Msn2/4 and removing repressors of the response (e.g., Sko1) (Proft & Struhl, 2002; Capaldi *et al*, 2008; Gutin *et al*, 2015). Another example is the cAMP phosphodiesterase (Pde1, Pde2) affecting response duration. The removal of these enzymes changes the rate of cAMP clearance (Ma *et al*, 1999), affecting the global timing of cellular decisions. An additional example is *GAL11* gene deletion, reducing transcription rate. Most likely this effect is at the promoter level: Gal11 is a component of the Mediator complex (Li *et al*, 1995) and has been shown to be on the Msn2-Mediator interface (Lallet *et al*, 2006; Sadeh *et al*, 2012). Thus, we assume that the absence of Gal11 affects the efficiency of establishing pre-initiation polymerase complex.

Some of the other results are more surprising. For example, mutations in elongator complex (Elp4, Elp6) and chromatin factors (Eaf7, Gcn5) also affect response duration. Their activity might be related to eviction of Msn2/4 from promoters, or change the global state of the cell. Another example is the *UMP1* gene, encoding for a chaperon involved in proteasome maturation, whose deletion reduces production rate. This factor was previously found to affect the nuclear degradation of Msn2, although under different conditions (Erkina *et al*, 2008; Sadeh *et al*, 2011). However, the observed effect might be due to complex indirect effects of proteasome imbalance. These observations require further scrutiny to be properly validated and understood.

### Finer analysis of genetic interactions

We reproduced an earlier double-mutant screen (Gutin *et al*, 2015), by assaying not only the end effect but also the response dynamics (Fig 3). Our analysis demonstrates the utility of this multi-dimensional phenotype. In particular, we are able to detect epistatic interactions that were obscured when focusing only on total response. Moreover, the epistatic interactions of specific phenotypes provided insights as to the source of interactions, thus simplifying the mechanisms that drive seemingly complex genetic interactions. Additionally, we were able to detect hundreds of epistatic relationships that were not detected in earlier analysis.

These epistatic interactions suggest pathway structure and potential underlying mechanisms. We explored a few of these here in detail. However, many of the others merit further exploration (Dataset EV2).

### Regulatory programs involved in establishing and extending the transient response to acute stress

Our results uncovered roles of Mck1 and Msn5 in modulating Msn2/4 activity. We observe two phases of Msn2/4 activity—an initial, acute, and synchronized phase followed by an extended progression phase which is necessary for induction of late responding genes. As we show, neither Mck1 nor Msn5 is necessary for the initial phase of Msn2/4 activity. However, export of Msn2/4 out of the nucleus by Msn5 and "licensing" of additional cycles of nuclear import, in an Mck1-dependent manner, are necessary for the second phase.

Earlier work dissected how different promoters respond to different regimes of Msn2 nuclear localization (Hansen & O'Shea, 2013, 2015). In particular, they suggest that some promoters are insensitive to short nuclear pulses and require an extended pulse for activation. Other promoters are more sensitive to the total amount of Msn2 in the nucleus, regardless of the dynamics of the pulses. Here, we observe a composition of two regimes, extended pulse followed by rapid succession of short pulses. Such a composition can attenuate the response of promoters, as we observe when we perturb the second phase (Fig 4). These observations raise the question of what regulatory strategies can be achieved by such compositions and whether they allow finer tuned regulation that can be achieved by single-phase response.

This multi-phase response provides a partial answer to some of the questions we set out to answer. When a cell is subjected to acute change in the environment, it has to respond quickly, hence the need for an acute response in a short time frame. Such a response,

however, cannot be calibrated to the intensity of the environmental insult. Indeed, the initial phase of Msn2/4 activation is a binary switch. Such an "all or nothing" response has a price: stress response halts cell cycle progression and cellular growth, and furthermore, it incurs heavy cost in production of stress-related proteins (Brauer *et al*, 2008; López-Maury *et al*, 2008; de Nadal *et al*, 2011). Thus, the early termination of the acute phase, well before the maturation of proteins expressed during this phase, limits the extent of the response. During this initial response interval, other cellular pathways can sense the extent of the insult and determine whether to prolong the response. These decisions determine the duration of the second phase. Indeed, this second phase is much more variable among cells, which can be related to other global aspects of the cell, such as its size or stage in the cell cycle, that influence the decision between extending the response or resuming growth. This flexibility increases the fitness of the population, as it avoids the costs incurred by "one size fits all" response which can be insufficient for some cells, and too much for others. Further scrutiny is needed to identify the mechanisms involved in these decisions, what are their inputs, and how do they integrate them.

# Materials and Methods

## Yeast strains

Yeast strains used in this study are listed in Table EV1. The Hsp12-GFP fusion strain was generated by genomic integration of PCR fragment amplified from Hsp12-GFP strain from the yeast GFP collection (Huh *et al*, 2003) to the query strain YMS140α. All single- and double-mutant strains (Figs 1–4) were adapted from Gutin *et al* (2015). Single-mutant strains used for Msn2 localization analysis (Fig 5) were generated by replacing the original ORFs in BY4741 strain with *URA3* marker followed by transformation of the centromeric plasmid containing the *MSN2-GFP* fusion under the control of a constitutive *ADH1* promoter and a *LEU2* marker (Görner *et al*, 1998). As a proxy for WT, that underwent the same genetic manipulation and has the same marker genes, we used Δhis3, which deletes a gene that is inactivated in the parent strain.

## FACS measurement of reporter gene dynamics

Yeast strains expressing Hsp12-GFP were grown in SC media at 30°C with constant shaking to OD < 1. In the single-mutant and double-mutant screens (Figs 2 and 3), cells were grown to mid-log in 96-well plates using a custom-designed robotic growth protocol (Gutin *et al*, 2015). At the beginning of the experiment, the cultures were mixed with SC + KCl (1.6/3.2 M) to reach the desired final KCl concentration and transferred to 96-well plates.

All plates were analyzed by high-throughput flow cytometry (BD FACSCalibur with CyTek upgrade) using the HyperCyt automated sampler (IntelliCyt). The plates were sampled in 6 time points with intervals of 18 min. The data of each plate were partitioned into individual wells and gated to remove dead cells, cell debris, and other non-typical events as described before (Gutin *et al*, 2015). The median fluorescence of the cells was calculated and corrected for autofluorescence by subtracting the median value of a strain without GFP tag.

## Parametric decomposition of stress dynamics

We used a simplified kinetic model to represent the dynamics of a transient transcriptional response (Fig 1C). Briefly, the transcriptional window starts at $t_{on}$ and ends at $t_{off}$. During this time interval, there is transcription with a constant rate $\alpha$. The mRNA molecules are degraded with an exponential degradation rate $\beta$ and translated to protein molecules with a constant rate. We assume that the mRNA degradation rate is constant between the different conditions and mutants. See Weiner *et al* (2012) for further details.

The computer code for extracting the parameters is provided as Computer Code EV1. To find the best set of parameters for each time course, we used MATLAB function "fmincon" using the "active-set" optimization algorithm. For each parameter, the median value over 2–3 biological repeats (Fig EV3A) was taken for further analysis.

## RNA-seq and analysis

### Growth and fixation

Yeast strains were grown in SC media at 30°C with constant shaking to OD < 1. Cells were then exposed to osmotic stress (final concentration of 0.4 M KCl) and fixed in cold methanol (−80°C) at 0, 5, 10, 15, 20, 30, 45, and 60 min after stress addition.

### RNA purification

RNA was purified using a high-throughput RNA isolation protocol (Dye *et al*, 2005). Briefly, cells were washed in ddw and incubated with Proteinase K (Epicentre MPRK092) and 1% SDS at 70°C to release the RNA. Cell debris was precipitated by centrifugation in the presence of KOAc precipitation solution. Finally, the RNA was purified from the supernatant using nucleic acid binding plates (UNIFILTER plates, catalog #7700-2810) and was stored with RNAse inhibitor (Murine #M0314L) at −80°C.

### 3′-RNA library preparation

RNA libraries were prepared as previously described (Klein-Brill *et al*, 2019). Briefly, RNA was reverse-transcribed using SmartScribe enzyme (TaKaRa) and in the presence of oligo-dT RT primers with a 7 bp barcode and a 8 bp UMI (a gift from Ido Amit). Barcoded samples were then pooled and purified using SPRI beads X1.2 (AMPure XP). DNA–RNA molecules were tagmented using Tn5 transposase (loaded with oligo Tn5MEDS-A) and 0.2% SDS was used to strip off the Tn5 from the DNA (Picelli *et al*, 2014) followed by a SPRI X2 cleanup. NGS sequences were added to the tagmented DNA by PCR (KAPA HiFi HotStart ReadyMix 2X-KAPA 33 Biosystems KM2605, 12 cycles). Finally, DNA was purified using double SPRI (X0.65 SPRI to clean large fragments followed by X0.8 SPRI beads).

## Sequencing and analysis

The library was paired-end sequenced using Illumina NextSeq-500 sequencer. Reads were mapped to the yeast genome (sacCer3) using bowtie2 with default parameters. Duplicated reads were filtered using UMI, to remove PCR bias. To estimate the expression level of each gene, we counted the number of reads that mapped to the 3′ end of the gene (from 350 bp upstream to 200 bp downstream of

TTS). The read counts in each sample were normalized to PPM (divided by the total number of reads and multiplied by $10^6$).

### Msn2/4-dependent production

We calculated the difference in RNA levels between each two consecutive time points in each strain. This value approximates the amount of RNA produced during this interval (Fig EV4A). Then, the difference profile of Δmsn2Δmsn4 was subtracted from the difference profile of the strain of interest (Δmck1, Δmsn5, WT), to capture an upper limit on the contribution of Msn2/4-dependent transcription in this strain. In the WT strain, this upper limit is by definition the Msn2/4-dependent contribution. On the background of other deletions, there might be synergistic effects between Msn2/4 and the deleted genes. However, when there is epistasis of Msn2/4 over the deleted gene, the quantity we report is the precise Msn2/4-dependent contribution.

### Time-lapse microscopy

For Hsp12-GFP accumulation analysis, mutant strains were grown in 96-well plates using the robotic growth protocol (Gutin *et al*, 2015). For Msn2-GFP localization experiment, the strains were grown in SC media at 30°C with constant shaking to OD < 1. The cells were then transferred to glass-bottom plate (384 format, Matrical Biosciences) coated with concanavalin A. The cells were left to descend to the bottom of the plate for 25 min and then gently washed to remove cells not attached to the glass. The media was replaced to SC + 0.4 M KCl at the beginning of the measurements. Time-lapse microscopy of the cells in bright field and GFP channels was taken using a scan-R high-content screening microscope (Olympus).

Image analysis was done using in-lab developed MATLAB program. Briefly, cell borders were identified using the bright field images. Cells in consecutive images were paired based on a reciprocal closest hit procedure. Cells that were traced throughout the whole time course were used in further analysis. For Hsp12-GFP accumulation analysis, average GFP levels were calculated for every cell and time point. For Msn2-GFP localization experiment, cells were tested for nuclear localization of the GFP reporter. Briefly, potential nuclei were found using object detection on the GFP channel within the cell borders, and filtered based on the ratio between the cell and the nucleus size.

### Epistasis analysis

We tested for epistasis interactions in the following parameters: total GFP produced, production rate, and response duration. Briefly, an epistasis interaction was called when the combination of two mutants with different effects on the parameter resembled the effect of one of the individual mutants. The test was performed as described previously in Gutin *et al* (2015).

## Data availability

The datasets produced in this study are available in the following databases:

- RNA-seq data: Gene Expression Omnibus GSE127851 (https://www.ncbi.nlm.nih.gov/geo/query/acc.cgi?acc=GSE127851).
- Code for data analysis : Computer Code EV1.

**Expanded View** for this article is available online.

## Acknowledgements

We thank A. Appleboim, D. Engelberg, G. Fialkoff, O. Rando, and members of the Friedman laboratory for comments and discussion. This work was supported by ERC AdG Grant ChromatinSys (340712), Israel Science Foundation I-CORE on Chromatin and RNA in Gene Regulation (1796/12), NIH grant RM1-HG006193.

## Author contributions

Conceptualization: JG and NF; Methodology and Investigation: JG and NF; Software and Formal Analysis: JG; Resources: JG, DJ-S, ES and AS; Writing and Visualization: NF and JG; and Supervision and Funding Acquisition: NF.

## Conflict of interest

The authors declare that they have no conflict of interest.

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
