## [Review Process File · Molecular Systems Biology]

Genetic screen of the yeast environmental stress response dynamics uncovers distinct regulatory phases

Jenia Gutin, Daphna Joseph-Strauss, Amit Sadeh, Eli Shalom and Nir Friedman.

Review timeline:	Submission date:	7 th April 2019
	Editorial Decision:	26 th May 2019
	Revision received:	21 st June 2019
	Editorial Decision:	18 th July 2019
	Revision received:	21 st July 2019
	Accepted:	29 th July 2019

Editor: Jingyi Hou

Transaction Report:

1st Editorial Decision

26th May 2019

Thank you again for submitting your work to Molecular Systems Biology. We have now received all the reports and as you will see below, the reviewers think that the presented findings seem interesting. They raise however a series of concerns, which we would kindly ask you to convincingly address in a revision. The recommendations provided by the reviewers are very clear and refer to further clarifications and some additional data analyses. Please feel free to contact me in case you would like to discuss in further detail any of the issues raised by the reviewers.

REFeree REPORTS

Reviewer #1:

Summary

Cells respond to environmental signals by inducing dozens or hundreds of genes, many of which are induced with different dynamics. To understand the types of regulation that can achieve the observed kinetics, and to identify the genes involved, Gutin et al. did a very impressive high-throughput flow-cytometry and RNA-seq timecourse to understand the kinetics of the yeast transcriptional response to salt stress.

To get a grip on these data they fit a model of gene expression, and thought about their results through the lens of the five parameters, three of which they focus on. This was very clever, as it allows us to think about the data in biologically and mechanistically relevant terms.

Epistasis analysis of induction kinetics in double mutants revealed, for example, that Hog1 is

dominant with regards to production rate, but not total open (induced transcription) time, while the exact opposite is true for Mck1.

The authors find that mck1 cells have a small defect in the initial wave of transcription, but the second wave is completely gone; this is likely the reason for the much shortened open time in mck1 cells. They then go on to show that mck1 cells have no defect in the initial round of Msn2/Msn4 nuclear localization, but fail to induce the second round.

The speculation is that, by having multiple genetically separable parameters (time to activation, length of activation, production rate) evolution is given greater freedom to tune a transcriptional response.

General remarks.

I am almost entirely convinced of the key conclusions. Before being 100% convinced I'd like to see a quantification of how well the model fits the data from all the genotypes and conditions, and how a control (non Msn2/Msn4 and/or non-stressed induced, ideally both) GFP reporter behaves.

Major points:

Transcriptional induction after KCl comes in two phases (eg, PMC3681706 and others) and that the stress response is finished by 60 minutes. When you make the point that constant protein levels imply that transcription stops, cite PMID10217506 (and maybe others, not my field) and make it more clear that the shutoff of transcription under mild stress is already known.

How much of the result in Figure 1A, 1B & 1D is due to non-STRE specific changes in expression? I'd like to see some induction kinetics of a control GFP (TEFpr-GFP, or perhaps better, an Msn2/Msn4 independent KCl induced gene). Is there any change in expression? I feel that this is important in interpreting, for example, the lag at high KCl. Is this STRE-specific or part of a global effect?

Define DeltaGFP, BasalGFP & T0 (Fig2A).

The model in Fig1 is wrong, as you point out later in the text. There are two transcriptional waves, and they probably have different production rates. Too many free parameters would probably cause the model to lose power, but please discuss this discrepancy in the discussion. How much / what kind of data, would you need to fit a two-wave model? Do you think a two-wave model is correct and/or necessary to understand the data?

How well does the model fit the data? What is the % of explainable variance (taking into account variation between replicates) you can predict? Or how well can a model train on two replicates predict expression dynamics of the third? S3A is good, but I'd like to have a better feel for how well the very simple single-production-rate model fits the data.

Is it valid to calculate Msn2/4 dependent production for an mck1 strain just using the mck1, WT, and msn2/msn4 data, without the mck1,msn2,msn4 triple mutant data? It's not clear to me that this is correct. If you compare HSP12-GFP expression kinetics in WT, msn2/4, mck1, and the triple, do you get the same value for Msn2/4 dependent expression no matter how you calculate it?

Please show the HSP12-GFP traces kinetics for mck1 and msn2,msn4 strains. These play an important role in the paper, yet you don't show the data.

Please include a summary with the number of replicates for each genotype for the RNAseq. I believe there is just one replicate for the mutant genotypes. True? In this case, are any of the differences at individual timepoints an individual gene significant (95% CI by bootstrap)? The Msn2 localization data and epistasis data are consistent with the RNA-seq results. I assume the HSP12-GFP are too, but I didn't see them. But most of figure 4 depends on a single replicate, and this should be more transparent.

Minor points

Fig2a: lowercase italics w/delta so we know these are data of deletions, not expression of genes

Fig 1A,B: need y axis ticks

Change 'Msn2/4 dependent production' to Msn2/4 dependent mRNA levels. To make it more clear that it's not protein, and you're measuring steady-state, not production.

The model has one transcription rate, but we know, from this paper and several that came before, that there are two transcription stages. This discordance should be discussed.

Fig1: Ton & Toff are a bit confusing. I think that Ton is the Time-to-ON, while Toff is the total time from stress to the shutoff of transcription. This could be made clearer in Fig1C. But is Toff really interesting? Why not just report Topen (which should be marked in Fig1C).

Fig4C , 4F (and others): It's not clear what the units are. UMI counts per million / UMI counts per million at t0? Please make this more clear.

mRNA half-lives are 5-20 minutes, so presumably "mRNA production" represents a mix of production and decay. "Production (difference from previous timepoint)" is ok.

Does "B. Similar to 4C" mean "Same data as in 4C"? If so, please write "same data" to differentiate it from a different replicate.

Stylistic (entirely optional) points:

It is only while writing this review that I fully understood the story. The authors generated a lot of data, and feel the need to present all of it. I feel that excess panels obscure the story and the novel findings. A culling of half the panels would make the story more clear.

Many figures take too much thought to figure out, while others are irrelevant to the main story line. Some examples:

Figures 3E,3F, 4A,4B: I believe the point of these is one of dominance. It's not the absolute values for Topen & alpha, but the difference of the double to the hog1/mck1 single. And the contrast between hog1 and mck1. Plotting the difference to the hog1/mck1 singles, or the calculated epistasis, might make this point more clear.

Figure 2A: Mandatory: What are deltaGFP, basalGFP and T0? I don't see how they're calculated. Optional: Maybe move to supplement if you don't use them. In the methods section you write: "time to activation, length of activation, production rate" - these are very clear. "Total open time" assumes chromatin architecture changes, and should be changed to a label that better reflects the data you've collected.

To me, the title "uncovering a succession of regulatory phases" implies that the two phases were not previously known. They were. Be more explicit as to which finding about the two phases are truly novel.

I sign my reviews

Lucas Carey

Center for Quantitative Biology

Peking-Tsinghua Center for Life Sciences

Academy for Advanced Interdisciplinary Studies

Peking University, Beijing

Reviewer #2:

Summary

Gutin et.al developed an elegant framework for studying the underlying genetic and temporal transcriptional response of yeast to osmotic shock, mimicking an induced environmental stress responses. The authors attempt to address the diverse requirements for a yeast stress response - a predetermined program following a stress shock or an adaptive response built on the duration and intensity of the stress - all at the same time incurring a fitness cost for a needlessly length response. By monitoring the induced HSP12-GFP protein levels (an indicator of stress response) on a per cell basis at multiple time points post stress induction, the authors could parameterize the stress response phenotype into three independent values - time on, off and rate of GFP production. Mapping the responses on this 3-parameter space in single and double deletion genetic backgrounds, the authors could effectively compare the genetic and environmental effects on the transcriptional regulatory pathways. The authors looked for epistatic interactions that extended the response in high stress conditions. They identified a central role of *mck1* and *msn5* gene coordinating *msn2/msn4* response. The authors conclude and posit a model of two separate regulatory phases involving an initial shuttle of active *msn2* into the nucleus and a slower shorter pulses of nuclear localization during stress progression caused by temporal changes in transcription of *mck1* and *msn5*.

General Remarks

The framework of parameterizing the induction of HSP12-GFP curve following stress is a very simple and elegant method for capturing the subtleties of the response. It places all experiments in a manner where comparisons between conditions or genetic backgrounds can be easily and accurately made. The authors provide strong mathematical evidence on the independence of the parameters and perform orthogonal measurements to resolve the phenotype of transcriptional stress response dynamics more finely into three independent parameter. A number of epistatic interactions were identified by this method and is an insightful method for additional comparisons. This is the very first time I have seen the subtle transcriptional changes in stress response be teased apart in such a beautiful manner.

The methodology introduced is scalable by incorporating other reporter proteins, more diverse genetic backgrounds and even conditions. The epistatic interaction data for the standard 0.4M KCl conditions will be a novel resource for groups focusing on yeast stress pathway and also broadly for researchers looking for or confirming candidates in their interaction studies. The writing is very clear, the problem has been well stated and the conclusions have been well backed up in general. A piece of the final study that involves shuttling of *msn2* in phases to explain the two phases need a few orthogonal evidences to build a stronger case for the existence of secondary transcriptional wave.

Major points

*Variance of parameters and measurements: The median measurement has been calculated in the FACs assay. The variance of these GFP measurements has not been described well, leading to difficulty in interpreting the sensitivity of the parameters and therefore the significance of an epistatic interaction. As an example - while looking at figure 3F, 4A or S3B, it is not clear why amongst closely spaced measurements, one mutant is called epistatic (pink) over the other (blue), what the cutoff was and how well are the parameters bounded to be called "same" effect. While not a fan of thresholds, I understand the need for demarcating biologically significant effect. My worry is a poor decision of a threshold, especially when serving as a community resource, may help reinforce a false positive observation. While the parameter estimates from the single cell studies seem concordant with population studies, it is still unclear to me the impact of the variability on the three parameters from the FACs assay alone. I would suggest the following -

- (1) Provide the distribution of the calculated GFP values for each time point in the FACs assay (as a boxplot; each point is a cell's GFP value) for the 0.4M KCl WT sample as a supplementary.
- (2) Clarify the method used for calling an epistatic interaction. Was the 20% +/- max GFP values (indicated in the cited paper) used for all the three parameters as well? Was this the criteria used in identifying the epistatic and non-epistatic interactors (pink vs blue dots) in the figures (indicated above). Indicating a dashed line for thresholded value around the single mutant grey line will be very informative for the reader.
- (3) If prior literature on the true epistatic interactions are well known or even a set annotated, it will be useful to perform sensitivity analysis by changing the values of the parameters by 20% and asking how many epistatic interactions were still recovered correctly.

* Evidence of activity in the progression phase: The proposed model for MSN2 shuttling between the nucleus and cytosol, induced by mck1 activity, in two phases is persuasive but would require additional evidence. It is fascinating that the stress response effect that appears to be purely post-translational (nucleus relocalizations) was captured by the transcriptional program and not by quicker changes such as phosphorylations by existing kinases that typically alter cellular conditions. Was this is due to the selection bias of examining genes regulating the total open time parameter and not production rates? The lines of evidence provided are the transcriptional changes in *msn2/4* in the different knockouts, mRNA changes in *mck1* and *msn5* deletions and the fusion protein shuttling in and out of the nucleus. While acknowledging that further mechanistic detail would be beyond the scope for this paper, direct evidence indicating that it is indeed a transcriptional led stress response will help the proposed model. I would suggest -

- (1) What is the effect of *mck1* deletion under different concentrations of KCl? Does a similar *mck1* expression occur in the progression phase? The FACS assay performed on extreme KCl concentrations (say 0.15 and 0.8M) and plotting it in the same manner as fig 4C will indicate the magnitude of the effect of *mck1* and the consistent observation of a secondary wave (in the higher concentration case).
- (2) Additionally transcript and protein levels of *mck1* (say by western blot) indicating that the levels does increase with the progression phase will be a direct evidence.
- (3) The figure 5A (panel B) is difficult to interpret. A supplementary time lapse video showing the MSN2-GFP shuttling in and out in the WT will help clarify what is being meant by the asynchronous periodic oscillations of *msn2*-GFP and further explain the graph. Do we see periods of nuclear GFP disappear or are these multiple GFP puncta moving in and out of nucleus or is asynchronous timing across cells?

Minor points

Explain the Delta and basal GFP in legend of figure 2
Scale bar on fig 2H.

Reviewer #3:

In this study, Gutin et al track the mRNA and protein production dynamics in yeast responding to environmental stress caused by KCl. More specifically, using the accumulation dynamics of Hsp12-GFP as their reporter for Msn2/4 activity in response to stress, the authors show that both the intensity and the duration of the response can be modulated. They simplify the presentation of the results by parametrization, and also show that population level data reasonably matches with the data obtained through single-cell microscopy experiments. Performing epistasis analysis on data obtained from strains missing one or two genes let the authors identify epistatic interactions between several genes. Interestingly, they find that Mck1 affects response duration through a secondary transcriptional wave. Finally, the authors report the observation of two distinct phases of Msn2/4 activity during stress-response.

I enjoyed reading this manuscript. This is a very interesting study whose novel results provide additional insights into our understanding of how cells modulate their response to environmental stress. The manuscript was well-written, and I found the experimental design to be comprehensive and elegantly implemented.

I have the following minor points to raise about this work:

1. The FACS experiments quantify total fluorescence from single cells, while for the microscopy experiments "average GFP levels were calculated for every cell and time point" according what is written in the methods section. The authors should discuss whether or not using these different metrics (total and average) would lead to FACS-microscopy-matching issues when differential KCl concentrations lead to cell-volume changes.
2. The authors state that "While the initial nuclear import is by and large a long continuous event (one entry and one exit), the later nuclear imports are in shorter pulses of multiple entries (Figures 5A, lower panel, and S5B)."

Following this statement, the authors should provide a potential explanation for how such a dynamic activity/regulation could be beneficial from an evolutionary point of view.

Please see attached detailed point-by-point response.
All changes in the manuscript are marked in red.

Reviewer #1:*Summary*

Cells respond to environmental signals by inducing dozens or hundreds of genes, many of which are induced with different dynamics. To understand the types of regulation that can achieve the observed kinetics, and to identify the genes involved, Gutin et al. did a very impressive high-throughput flow-cytometry and RNA-seq timecourse to understand the kinetics of the yeast transcriptional response to salt stress.

To get a grip on these data they fit a model of gene expression, and thought about their results through the lens of the five parameters, three of which they focus on. This was very clever, as it allows us to think about the data in biologically and mechanistically relevant terms.

Epistasis analysis of induction kinetics in double mutants revealed, for example, that Hog1 is dominant with regards to production rate, but not total open (induced transcription) time, while the exact opposite is true for Mck1.

The authors find that mck1 cells have a small defect in the initial wave of transcription, but the second wave is completely gone; this is likely the reason for the much shortened open time in mck1 cells. They then go on to show that mck1 cells have no defect in the initial round of Msn2/Msn4 nuclear localization, but fail to induce the second round.

The speculation is that, by having multiple genetically separable parameters (time to activation, length of activation, production rate) evolution is given greater freedom to tune a transcriptional response.

General remarks.

I am almost entirely convinced of the key conclusions. Before being 100% convinced I'd like to see a quantification of how well the model fits the data from all the genotypes and conditions, and how a control (non Msn2/Msn4 and/or non-stressed induced, ideally both) GFP reporter behaves.

We thank the reviewer for the positive comments.

Major points:

Transcriptional induction after KCl comes in two phases (eg, PMC3681706 and others) and that the stress response is finished by 60 minutes. When you make the point that constant protein levels imply that transcription stops, cite PMID10217506 (and maybe others, not my field) and make it more clear that the shutoff of transcription under mild stress is already known.

We thank the reviewer and modified the text (page 4, second paragraph of the results section) to describe the known dynamics of yeast adaptation to osmotic stress. We reference a review paper that includes the specific result the reviewer mentions and many others.

How much of the result in Figure 1A, 1B & 1D is due to non-STRE specific changes in expression? I'd like to see some induction kinetics of a control GFP (TEFpr-GFP, or perhaps better, an Msn2/Msn4 independent KCl induced gene). Is there any change in expression? I feel that this is important in interpreting, for example, the lag at high KCl. Is this STRE-specific or part of a global effect?

The dynamics of yeast response to KCl levels has been extensively studied, which is the reason for choosing this model system. In particular, earlier works did comprehensive analysis of Msn2/4 dependence of different response genes (e.g., Gasch et al 2000, O'Rourke and Hershkovich 2004, Capaldi et al 2008). These works show that the lag is a general feature due to time needed to clear the nucleus of high salt concentration (Proft & Struhl 2004) and recover other aspects of cellular state (Muzzey et al 2009).

To illustrate this phenomena in non-STRE genes, we can examine response of STL1, the canonical MSN2/4 independent response gene, to different levels of salt:

Similar lag is seen in all other non-STRE induced genes. These results recapitulate previously published expression measurements, and thus are not included in the paper.

Define DeltaGFP, BasalGFP & T0 (Fig2A).

Fixed.

The model in Fig1 is wrong, as you point out later in the text. There are two transcriptional waves, and they probably have different production rates. Too many free parameters would probably cause the model to lose power, but please discuss this discrepancy in the discussion. How much / what kind of data, would you need to fit a two-wave model? Do you think a two-wave model is correct and/or necessary to understand the data?

As the (in)famous quote goes “all models are wrong, but some are useful”. In this case there is little difference in fit to the data between a model with two consecutive transcriptional waves and one with a single longer wave. The relevance of the second wave appears when we do genetic dissection to show that the second wave (e.g., the extension of the “single wave” in the model) depends on specific proteins. So to answer the reviewer’s question, we prefer to err on the side of a simpler model, and following in another (in)famous quote “everything should be made as simple as possible, but not simpler”, we believe that the model we use is the simplest model describing the phenomena.

How well does the model fit the data? What is the % of explainable variance (taking into account variation between replicates) you can predict? Or how well can a model train on two replicates predict expression dynamics of the third? S3A is good, but I'd like to have a better feel for how well the very simple single-production-rate model fits the data.

We thank the reviewer for the comment. As Figures 1A and B show, and as mentioned in the text the fit is very good in these examples. To show the generality of these good fits, we added supplemental Figure EV3B to show error for measurements in our screen (all strains X all time points) showing that the model explains > 99.9% of the variance. In addition, Figures EV1B and EV1C show, the fit is robust to downsampling of time points.

Is it valid to calculate Msn2/4 dependent production for an mck1 strain just using the mck1, WT, and msn2/msn4 data, without the mck1,msn2,msn4 triple mutant data? It's not clear to me that this is correct. If you compare HSP12-GFP expression kinetics in WT, msn2/4, mck1, and the triple, do you get the same value for Msn2/4 dependent expression no matter how you calculate it?

We thank the reviewer for raising this point. Indeed, as the reviewer points out, the quantity we report on is not formally the Msn2/4 dependent production for mck1 strain. However, if msn2/4 is epistatic over mck1, then this would have been the correct quantity. Given that in early response phase mck1 has essentially no effect (Figure 4C), and that mck1 affects the same genes as msn2/4 (Figure 4D), it is reasonable to assume that most, if not all, of the Mck1-dependent effect is through Msn2/4. Thus, we believe our estimate is relevant for the early stage, and in later stages, our estimate is a lower-bound on the Msn2/4 dependent production. We clarified the issue in the text (bottom of page 9) and call the entity “approximate Msn2/4-dependent production”.

Please show the HSP12-GFP traces kinetics for mck1 and msn2,msn4 strains. These play an important role in the paper, yet you don't show the data.

We thank the reviewer for the comment. We added these in Supplementary Figure EV4A.

Please include a summary with the number of replicates for each genotype for the RNAseq. I believe there is just one replicate for the mutant genotypes. True? In this case, are any of the differences at individual timepoints an individual gene significant (95% CI by bootstrap)? The Msn2 localization data and epistasis data are consistent with the RNA-seq results. I assume the HSP12-GFP are too, but I didn't see them. But most of figure 4 depends on a single replicate, and this should be more transparent.

We modified the text to reflect the single replicate in most strains (page 6, paragraph 3). The WT was done in triplicate and hog1 was done twice. These repeats were consistent with each other.

Regarding significance, All the stress induced genes show significant changes when compared to pre-stress. The p-values of these changes are extremely low, and thus we consider reporting them meaningless. The differences between expression of the same gene between mutants vary in their extent. However, they are clearly separated from the bulk of genes that agree. For example, here we compare the RNA-seq counts for all genes 30 minutes after stress. On the left is WT rep1 (x-axis) compared to WT rep2 (y-axis), the middle graph shows WT (x-axis) vs mck1 (y-axis) and on the right is WT (x-axis) vs. msn2msn4 (y-axis). The points in red are the 230 stress induced genes we analyzed in Figure 4 (which were selected based on the WT time course).

Two observations. First, the bulk of the genes are on the diagonal, and so the effects we are reporting are specific to the stress induced genes. Second, most of the induced genes (the Msn2-dependent genes) are clearly off diagonal, much more so than the variation between samples in all other genes (and in repeats).

Since we are reporting on a phenomena that appears consistently in a large set of genes, we believe that adding a lot of numbers to supplementary tables will not be productive.

Minor points

Fig2a: lowercase italics w/delta so we know these are data of deletions, not expression of genes

Done

Fig 1A,B: need y axis ticks

Done

Change 'Msn2/4 dependent production' to Msn2/4 dependent mRNA levels. To make it more clear that it's not protein, and you're measuring steady-state, not production.

We are not sure what the reviewer is referring to here. As evident by the comment below (about mRNA half life), the reviewer does agree we are examining mRNA production when comparing consecutive time points. Thus, differences between WT and msn2/4 shows the Msn2/4-dependent production.

The model has one transcription rate, but we know, from this paper and several that came before, that there are two transcription stages. This discordance should be discussed.

See comment above regarding model "correctness"

Fig1: Ton & Toff are a bit confusing. I think that Ton is the Time-to-ON, while Toff is the total time from stress to the shutoff of transcription. This could be made clearer in Fig1C. But is Toff really interesting? Why not just report Topen (which should be marked in Fig1C).

We clarified the definition of the parameters in Figure 1C and defined Topen in the text (page 5, paragraph 4). There is a debate which one is more natural - Toff is the one we estimate in the model, so we present it in Figure 1D. However, Starting from Figure 1F we focus only on Topen, and we thank the reviewer for suggesting to give a name and clearer definition.

Fig4C , 4F (and others): It's not clear what the units are. UMI counts per million / UMI counts per million at t0? Please make this more clear.

We thank the reviewer for the comment. We adjusted all relevant figures (2,4,5,EV4,EV5) and now the units are RPM (after UMI based filtration).

mRNA half-lives are 5-20 minutes, so presumably "mRNA production" represents a mix of production and decay. "Production (difference from previous timepoint)" is ok.

We thank the reviewer for the comment. We adjusted figures 4,5,EV4,EV5 accordingly.

Does "B. Similar to 4C" mean "Same data as in 4C"? If so, please write "same data" to differentiate it from a different replicate.

Done

Stylistic (entirely optional) points:

It is only while writing this review that I fully understood the story. The authors generated a lot of data, and feel the need to present all of it. I feel that excess panels obscure the story and the novel findings. A culling of half the panels would make the story more clear.

We appreciate the comment and tried to take this spirit into account while revisiting the manuscript.

Many figures take too much thought to figure out, while others are irrelevant to the main story line. Some examples:

Figures 3E,3F, 4A,4B: I believe the point of these is one of dominance. It's not the absolute values for Topen & alpha, but the difference of the double to the hog1/mck1 single. And the contrast between hog1 and mck1. Plotting the difference to the hog1/mck1 singles, or the calculated epistasis, might make this point more clear.

Figure 2A: Mandatory: What are deltaGFP, basalGFP and T0? I don't see how they're calculated. Optional: Maybe move to supplement if you don't use them. In the methods section you write: "time to activation, length of activation, production rate" - these are very clear. "Total open time" assumes chromatin architecture changes, and should be changed to a label that better reflects the data you've collected.

We thank the reviewer for the comment, we clarified the definition of the parameters (page 19, caption for Figure 2A).

To me, the title "uncovers a succession of regulatory phases" implies that the two phases were not previously known. They were. Be more explicit as to which finding about the two phases are truly novel.

We appreciate the comment. The novel finding is the distinct regulation of the two phases. We changed the title accordingly (page 1, title).

*I sign my reviews
Lucas Carey
Center for Quantitative Biology
Peking-Tsinghua Center for Life Sciences
Academy for Advanced Interdisciplinary Studies
Peking University, Beijing*

Reviewer #2:

Summary

*Gutin et.al developed an elegant framework for studying the underlying genetic and temporal transcriptional response of yeast to osmotic shock, mimicking an induced environmental stress responses. The authors attempt to address the diverse requirements for a yeast stress response - a predetermined program following a stress shock or an adaptive response built on the duration and intensity of the stress - all at the same time incurring a fitness cost for a needlessly length response. By monitoring the induced HSP12-GFP protein levels (an indicator of stress response) on a per cell basis at multiple time points post stress induction, the authors could parameterize the stress response phenotype into three independent values - time on, off and rate of GFP production. Mapping the responses on this 3-parameter space in single and double deletion genetic backgrounds, the authors could effectively compare the genetic and environmental effects on the transcriptional regulatory pathways. The authors looked for epistatic interactions that extended the response in high stress conditions. They identified a central role of *mck1* and *msn5* gene coordinating *msn2/msn4* response. The authors conclude and posit a model of two separate regulatory phases involving an initial shuttle of active *msn2* into the nucleus and a slower shorter pulses of nuclear localization during stress progression caused by temporal changes in transcription of *mck1* and *msn5*.*

General Remarks

The framework of parameterizing the induction of HSP12-GFP curve following stress is a very simple and elegant method for capturing the subtleties of the response. It places all experiments in a manner where comparisons between conditions or genetic backgrounds can be easily and accurately made. The authors provide strong mathematical evidence on the independence of the parameters and perform orthogonal measurements to resolve the phenotype of transcriptional stress response dynamics more finely into three independent parameter. A number of epistatic interactions were identified by this method and is an insightful method for additional comparisons. This is the very first time I have seen the subtle transcriptional changes in stress response be teased apart in such a beautiful manner.

*The methodology introduced is scalable by incorporating other reporter proteins, more diverse genetic backgrounds and even conditions. The epistatic interaction data for the standard 0.4M KCl conditions will be a novel resource for groups focusing on yeast stress pathway and also broadly for researchers looking for or confirming candidates in their interaction studies. The writing is very clear, the problem has been well stated and the conclusions have been well backed up in general. A piece of the final study that involves shuttling of *msn2* in phases to explain the two phases need a few orthogonal evidences to build a stronger case for the existence of secondary transcriptional wave.*

We thank the reviewer for these kind words.

Major points

**Variance of parameters and measurements: The median measurement has been calculated in the FACs assay. The variance of these GFP measurements has not been described well, leading to difficulty in interpreting the sensitivity of the parameters and therefore the significance of an epistatic interaction. As an example - while looking at figure 3F, 4A or S3B, it is not clear why amongst closely spaced measurements, one mutant is called epistatic (pink) over the other (blue), what the cutoff was and how well are the parameters bounded to be called "same" effect. While not a fan of thresholds, I understand the need for demarcating biologically significant effect. My worry is a poor decision of a threshold, especially when serving as a community resource, may help reinforce a false positive observation. While the parameter estimates from the single cell studies seem concordant with population studies, it is still unclear to me the impact of the variability on the three parameters from the FACs assay alone. I would suggest the following -*

We thank the reviewer for raising this issue. We are discussing it following reviewer #1 question (starting with: "How well does the model fit the data?"). There are several supplementary figures referring to the quality of fit and its robustness and the reproducibility of the estimation.

(1) Provide the distribution of the calculated GFP values for each time point in the FACs assay (as a boxplot; each point is a cell's GFP value) for the 0.4M KCl WT sample as a supplementary.

We thank the reviewer for the suggestion. We added Figure EV1A.

(2) Clarify the method used for calling an epistatic interaction. Was the 20% +/- max GFP values (indicated in the cited paper) used for all the three parameters as well? Was this the criteria used in identifying the epistatic and non-epistatic interactors (pink vs blue dots) in the figures (indicated above). Indicating a dashed line for thresholded value around the single mutant grey line will be very informative for the reader.

We thank the reviewer for the comment. The test for epistasis we used is exactly the same as in the previous paper and was used for all three parameters. To clarify it we followed the suggestion of the reviewer and added a shaded gray area that marks the epistasis in the relevant figures (3E,3F,EV3C,4A,4B).

(3) If prior literature on the true epistatic interactions are well known or even a set annotated, it will be useful to perform sensitivity analysis by changing the values of the parameters by 20% and asking how many epistatic interactions were still recovered correctly.

We appreciate the desire to quantify the success of the epistasis test. However, compiling high confidence test set of interactions from previous literature is a non-trivial. Moreover, only a few works dealt with transcriptional response to stress, and insights from steady state (or growth) interactions might be misleading. Thus, we believe that such a compilation is beyond the scope of this paper.

Our main point in the epistasis analysis section is that examining the two parameters uncover relations that are obscured when using total GFP. This conclusion is true at a range of cutoffs. We view the definition of epistasis as a working tool to identify interactions of interest. We are not making stronger claims about the correctness of these relations over ones that fail to pass the test due to specific threshold.

** Evidence of activity in the progression phase: The proposed model for MSN2 shuttling between the nucleus and cytosol, induced by mck1 activity, in two phases is persuasive but would require additional evidence. It is fascinating that the stress response effect that appears to be purely post-translational (nucleus relocalizations) was captured by the transcriptional program and not by quicker changes such as phosphorylations by existing kinases that typically alter cellular conditions. Was this is due to the selection bias of examining genes regulating the total open time parameter and not production rates? The lines of evidence provided are the transcriptional changes in msn2/4 in the different knockouts, mRNA changes in mck1 and msn5 deletions and the fusion protein shuttling in and out of the nucleus. While acknowledging that further mechanistic detail would be beyond the scope for this paper, direct evidence indicating that it is indeed a transcriptional led stress response will help the proposed model.*

I would suggest -

(1) What is the effect of mck1 deletion under different concentrations of KCl? Does a similar mck1 expression occur in the progression phase? The FACs assay performed on extreme KCl concentrations (say 0.15 and 0.8M) and plotting it in the same manner as fig 4C will indicate the magnitude of the effect of mck1 and the consistent observation of a secondary wave (in the higher concentration case).

Our data, using HSP12-GFP reporter, shows that indeed mck1 deletion shortens the response at a range of KCl concentrations:

Top panels - HSP12-GFP in WT (left) and mck1 (right). Bottom panels - comparison of estimated parameters for total open time (left) and transcription rate (right). In the range 0.3-0.8 KCl we see that mck1 mainly shortens the response with little change to transcription rate. In 0.2KCl the variability in the population leads to different response curve.

(2) Additionally transcript and protein levels of mck1 (say by western blot) indicating that the levels does increase with the progression phase will be a direct evidence.

We do not claim that protein levels of Mck1 are changing during progression phase. In fact, our data shows that Mck1 mRNA levels are stable throughout the stress response:

(3) The figure 5A (panel B) is difficult to interpret. A supplementary time lapse video showing the MSN2-GFP shuttling in and out in the WT will help clarify what is being meant by the asynchronous periodic oscillations of *msn2*-GFP and further explain the graph. Do we see periods of nuclear GFP disappear or are these multiple GFP puncta moving in and out of nucleus or is asynchronous timing across cells?

We added a supplemental video showing time lapse of one field of WT cells. We also highlighted in Figure EV5B the two phases. As can be seen in that figure and in the video, the nuclear import is highly synchronized in the first response phase and asynchronous in the second phase.

Minor points

Explain the Delta and basal GFP in legend of figure 2

Done

Scale bar on fig 2H.

Done

Reviewer #3:

In this study, Gutin et al track the mRNA and protein production dynamics in yeast responding to environmental stress caused by KCl. More specifically, using the

accumulation dynamics of Hsp12-GFP as their reporter for Msn2/4 activity in response to stress, the authors show that both the intensity and the duration of the response can be modulated. They simplify the presentation of the results by parametrization, and also show that population level data reasonably matches with the data obtained through single-cell microscopy experiments. Performing epistasis analysis on data obtained from strains missing one or two genes let the authors identify epistatic interactions between several genes. Interestingly, they find that Mck1 affects response duration through a secondary transcriptional wave. Finally, the authors report the observation of two distinct phases of Msn2/4 activity during stress-response.

I enjoyed reading this manuscript. This is a very interesting study whose novel results provide additional insights into our understanding of how cells modulate their response to environmental stress. The manuscript was well-written, and I found the experimental design to be comprehensive and elegantly implemented.

We thank the reviewer for these kind words.

I have the following minor points to raise about this work:

1. The FACS experiments quantify total fluorescence from single cells, while for the microscopy experiments "average GFP levels were calculated for every cell and time point" according what is written in the methods section. The authors should discuss whether or not using these different metrics (total and average) would lead to FACS-microscopy-matching issues when differential KCl concentrations lead to cell-volume changes.

We thank the reviewer for this comment. There are two issues here - the comparison of two technologies, and the possible changes due to osmotic shock.

Regarding the first, the FACS signal we use is the signal height, which is maximal fluorescence readout for the cell. We argue what it corresponds better with average GFP microscopy, as the width of the FACS laser is larger than a pixel in the microscope image, and thus provide average over the cell. However, it is clear that this is not a perfect agreement.

Regarding the second issue --- all parameters, for both measurement devices, including total GFP in microscopy, change when cells shrink due to osmotic stress. Some change more than others. However, since the GFP levels increase after the actual stress activated transcriptional response, this is not a significant issue --- most cells regain their volume at this stage. The differences between different levels of KCl (e.g., Figure 1) are consistent also when using total GFP or GFP area.

2. The authors state that "While the initial nuclear import is by and large a long continuous event (one entry and one exit), the later nuclear imports are in shorter pulses of multiple entries (Figures 5A, lower panel, and S5B)."

Following this statement, the authors should provide a potential explanation for how such a dynamic activity/regulation could be beneficial from an evolutionary point of view.

We touch on this issue in the last paragraph of the discussion. Briefly, the variability of the second phase enables matching the individual stress response to the precise state of the cells (e.g., size, cell cycle phase). This flexibility increases the fitness of the population, as it avoids the costs incurred by "one size fits all" response (either too little for some cells, and too much for others). Following the suggestion of the reviewer we extended the discussion (page 16, first paragraph).

Thank you for sending us your revised manuscript. We have now heard back from the two reviewers who were asked to evaluate your revision. As you will see the reviewers are satisfied with the modifications made and think that the study is now suitable for publication. I am pleased to inform you that we will be able to accept your paper for publication pending the following minor amendments and editorial issues.

REFEREE REPORTS

Reviewer #1:

The authors have addressed all my concerns.

Reviewer #2:

The authors have sufficiently addressed a number of concerns by providing both additional data and clarification in the text. The two major concerns, I had - (a) sensitivity analysis concern of the model (also raised by other referees) has been supported by providing additional supplementary figures (EV1A) and information in the answers; (b) the mechanism of the *msn2* shuttling was also clarified to adequate extent by providing the movie and additional information that was suspected. In short, this manuscript has provided a very unique method for teasing apart subtle transcriptional regulation effect and make parametric comparison between genetic and conditional effects. I enthusiastically support the work for publication.

YOU MUST COMPLETE ALL CELLS WITH A PINK BACKGROUND ↓
PLEASE NOTE THAT THIS CHECKLIST WILL BE PUBLISHED ALONGSIDE YOUR PAPER

Corresponding Author Name: Nir Friedman

Manuscript Number: MSB-19-8939